# Reward recalibrates rule representations in human amygdala and hippocampus intracranial recordings

Luis Manssuer [1,2,3] ✉, Qiong Ding [2], Yashu Feng[3], Ruoqi Yang [3,4], Wei Liu[1], Bomin Sun[1], Shikun Zhan[1] ✉ & Valerie Voon[1,2,3] ✉

Adaptive behavior requires the ability to shift responding within (intra-dimensional) or between (extra-dimensional) stimulus dimensions when reward contingencies change. Studies of shifting in humans have focused mainly on the prefrontal cortex and/ or have been restricted to indirect measures of neural activity such as fMRI and lesions. Here, we demonstrate the importance of the amygdala and hippocampus by recording local field potentials directly from these regions intracranially in human epilepsy patients. Reward signals were coded in the high frequency gamma activity (HFG; 60-250 Hz) of both regions and synchronised via low frequency (3-5 Hz) phase-locking only after a shift when patients did not already know the rule and it signalled to stop shifting ("Win-Stay"). In contrast, HFG punishment signals were only seen in the amygdala when the rule then changed and it signalled to start shifting ("Lose-Shift"). During decision-making, hippocampal HFG was more inhibited on non-shift relative to shift trials, suggesting a role in preventing interference in rule representation and amygdala HFG was sensitive to stimulus novelty. The findings expand our understanding of human amygdala-hippocampal function and shifting processes, the disruption of which could contribute to shifting deficits in neuropsychiatric disorders.

The ability to form, evaluate and implement rules, particularly in complex and changing environments, is a cornerstone of adaptive behavior. The rules we use to guide behavior often must be changed when they are no longer beneficial through a process of trial and error and strategies such as win-stay-lose-shift. Neuroscientists have been particularly interested in understanding differences in switching between concrete and abstract rules which may be affected differently in different neuropsychiatric disorders and inform better suited treatments[1,2]. Concrete rules involve direct stimulus-response associations. For example, a visual stimulus might signal a juice reward will be received if a particular lever is pulled. In contrast, abstract rules are independent of the specific stimuli they are applied to and generalize

to novel exemplars as studied using the Wisconsin card sorting test (WCST) in which subjects must choose stimuli based on dimensions such as shape to the exclusion of other features such as color[3]. Shifts in concrete and abstract rules can be dissociated with rule changes within (intradimensional; ID) and between (extradimensional; ED) stimulus dimensions. Whereas ID shifts involve reversal learning, ED shifts involve executive functions to shift attention.

The study of the neural basis of ID and ED shifting has largely centered on the prefrontal cortex (PFC), particularly orbitofrontal (OFC) and ventral-lateral prefrontal (VLPFC) cortices, due to association with reversal learning and executive function, respectively[3-7]. In contrast, limbic regions, such as the amygdala and hippocampus, have

[1]Department of Neurosurgery, RuiJin Hospital, Shanghai Jiao Tong University School of Medicine, Shanghai, China. [2]Department of Psychiatry, Addenbrookes Hospital, University of Cambridge, Cambridge CB2 0QQ, United Kingdom. [3]Neural and Intelligence Engineering Center, Institute of Science and Technology for Brain-Inspired Intelligence, Fudan University, Shanghai, China. [4]Deceased: Ruoqi Yang. ✉e-mail: lrm48@cam.ac.uk; shikun_zhan@163.com; vv247@cam.ac.uk

received less attention. This may be partly because the amygdala was believed to be important for emotion, fear and social cognition and the hippocampus memory and navigation. However, single-unit recording work in primates and rodents have shown that these regions play a much broader role in cognition relevant to ED and ID shifting. The amygdala has been shown to code reward, learning, attention, novelty, value-based decision-making, and task set representation[8–14]. This has led to the idea of the amygdala as a multidimensional processor that integrates cognitive and emotional functions by way of its interactions with the extensive number of regions it has anatomical connections to, including PFC[15,16]. Similarly, the hippocampus is now believed to not only contain maps of the spatial environment but also more generally maps of task space – abstract representations of different features of a task such as stimuli, responses, and values, how they unfold in time and space and how they depend on each other[17,18]. Indeed, hippocampal neurons have now been shown to code different points in a value space determined by the relative reward values of a set of stimuli[19].

Unfortunately, we know little about the inner workings of the human amygdala and hippocampus, as studies are usually restricted to fMRI, which can suffer from signal dropout, does not measure neural activity directly, has poor temporal resolution and cannot differentiate excitatory from inhibitory activity. One way around these issues is to make use of intracranial EEG (IEEG) recordings from epilepsy patients who often have electrodes implanted in these regions to determine seizure onset zones for resection. Such recordings are a direct measure of neural activity and have excellent spatiotemporal resolution and signal to noise ratio. Different frequency bands of the LFP provide information about neural processes with distinct physiological generators. In particular, high frequency gamma (HFG; 60–250 Hz) activity is believed to reflect population level spiking and be coordinated by low frequency oscillations (1–30 Hz) related to post-synaptic potentials[20–23]. Unlike event-related potentials (ERPs) or low frequency oscillations, HFG can more clearly demonstrate both task-induced activation and suppressions[23] of neural activity, has been shown to respond with similar temporal dynamics and to encode similar information as single unit activity[24]. It has been used to demonstrate reward and decision-making information coding in many brain regions[24–29]. However, the role of HFG in the human amygdala and hippocampus in ED and ID shifting remains unknown. There is some evidence from single neuron recordings in macaques[30] and imaging in humans[31] that the amygdala is inflexible to rapid reversals in reward contingencies and that amygdala lesions improve reversal learning performance[32], and therefore that it codes reward contingencies over longer time periods[9]. This contrasts with the OFC which shows rapid reversal learning[7]. However, these studies measured responses to the discriminative stimuli or used tasks that may be solved without needing outcome information. It is an open question whether outcome responses in the amygdala and hippocampus play a more complex role in shifting than previously thought.

While there have been several IEEG studies of the amygdala and hippocampus, the majority have maintained a focus on emotional, social or memory functions. In this study, we combine IEEG recordings with a task that can more accurately dissociate ED and ID shifting processes relative to the WCST, to more precisely study the role of the human amygdala and hippocampus in decision-making and feedback processing involved in shifting. Using a combination of mass-uni-variate, machine learning and connectivity analyses, our findings show the amygdala-hippocampus network is involved in multiple processes important for rule switching. Amygdala responses in the HFG band to reward and punishment only occurred when they signaled that patients should change from a shifting to a staying strategy (win-stay) and vice versa (lose-shift), suggesting that its reinforcement learning functions can be determined by goals and attention. The hippocampus also showed HFG win-stay signals but was otherwise inhibited suggesting a role in updating the correct rule in memory and preventing

interference. The win-stay signals in both regions were integrated via phase locking between low frequency oscillations in the delta-theta (2–5 Hz) band. In contrast to previous studies which only investigated ID shifts, focused on PFC or used less informative measures such as fMRI or lesions, the win-stay signals were present for both ED and ID shift trials. Additional HFG and theta (3–6 Hz) signals related to novelty and decision-making, which would facilitate task performance were also found in the amygdala, consistent with the multidimensional coding model. The findings advance our understanding of human amygdala and hippocampal function and suggest revisions to our models of how the brain accomplishes shifting which may be useful for understanding and treating deficits in shifting that may contribute to impulsivity and compulsivity in disorders such as addiction and OCD.

## Results

### Behavior
Seventeen epilepsy patients completed an ED and ID shifting task. Local field potentials (LFPs) were recorded concurrently from depth electrodes implanted in the amygdala of 14 patients and hippocampus of 13 patients (Fig. 1A). In the EDID task, two distinct shapes were presented on either side of the screen (Fig. 1B). Letters were superimposed on top of the shapes. Letters and shapes constituted two stimulus dimensions. On each trial, patients had to find which one of the four stimuli was correct by pressing one of two buttons corresponding to its location on the screen and then monitoring for correct or incorrect feedback. Once they had selected the correct stimulus for three trials in a row, the rule would then change either within (ID) or between (ED) stimulus dimensions. For example, it might change from T to S (ID) or T to ellipse (ED). After six rule changes, a new stimulus set would appear. There were four stimulus sets in total. Two choices were made on each trial so that the superimposition of the letters and shapes could be swapped to allow for calculation of which stimulus the patient believed to be the rule. Unlike conventional WCST or reversal tasks, the type of shift made on each trial was determined by the difference in patients' choices between successive trials. This allowed us to decorrelate ED from ID shifting which is an issue in the WCST.

Across all analyses, we used the first choice that patients made as this is when they are making shifts and decisions. The second choice is simply a repetition of the first and therefore does not necessitate such processes – its only purpose was to allow us to measure which stimulus patients thought was the correct rule on each trial. The reaction times were analyzed using a linear mixed effects (LME) model with the fixed effects factors of shift type (ID, ED and no shift) and the random effects factor of patient. Shift type was determined by the patients' choice on each trial relative to the previous trial. The Bonferroni adjusted $p$-value threshold of .0167 was used to correct for the three comparisons made. One-tailed $p$-values were used based a priori on a previous study using a very similar EDID task which showed slower reaction times on shift trials relative to no shift trials and ED relative to ID trials[5]. Indeed, reaction times were faster on no shift trials relative to ED ($t(2056) = -7.8$, $p < .0001$, estimate = −.34, 95% CIs = [−.43 −.26]) and ID ($t(1873) = -3.5$, $p < .001$, estimate = −.082, 95% CIs = [−.13 −.036]) shift trials and ID shift trials were faster than ED trials ($t(1081) = 2.2$, $p = .008$, estimate = −.34, 95% CIs = [−.43 −.26]) (Fig. 2A). Very similar results were also found when conditions were defined by block type (See Supplementary Fig. 1). ID shift trials were faster than ED trials ($t(2238) = -4.4$, $p < .0001$, estimate = −.11, 95% CIs = [−.17 −.06]). Furthermore, and consistent with previous studies[5], there were slower reaction times on set change trials relative to ED ($t(1496) = -7.5$, $p < .0001$, estimate = −.22, 95% CIs = [−.28 −.16]) and ID ($t(1412) = -9.5$, $p < .0001$, estimate = −.28, 95% CIs = [−.33 −.22]) shift trials.

The numbers of errors in blocks requiring an ID and ED shift were compared with each other as well as with blocks in which the stimulus set changed and therefore did not necessitate an ID or ED. Errors were analyzed using a generalized linear mixed effects (GLME) model with

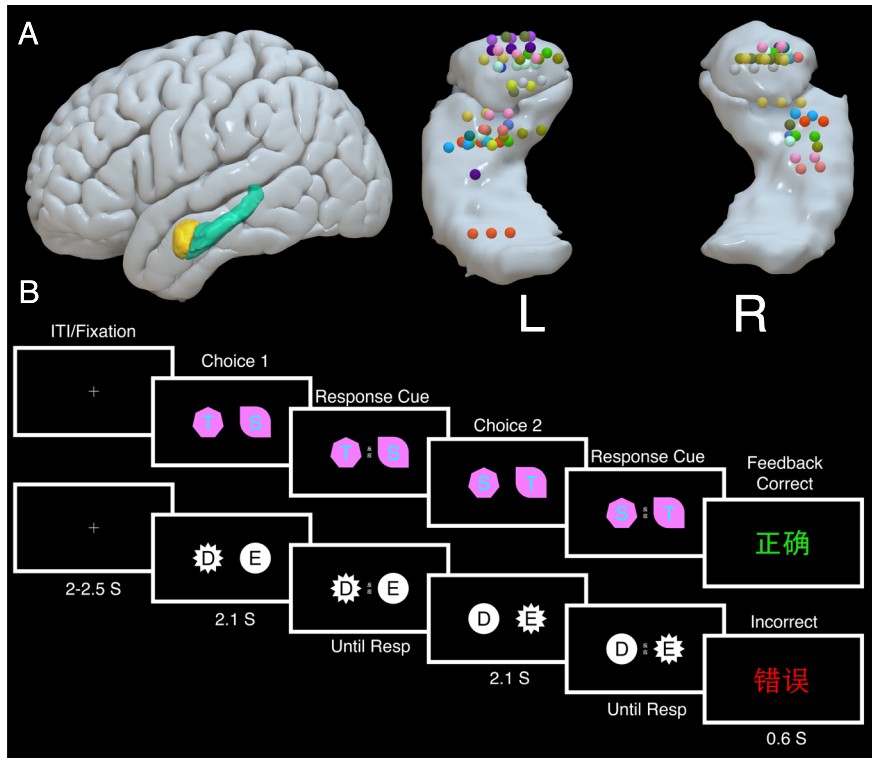

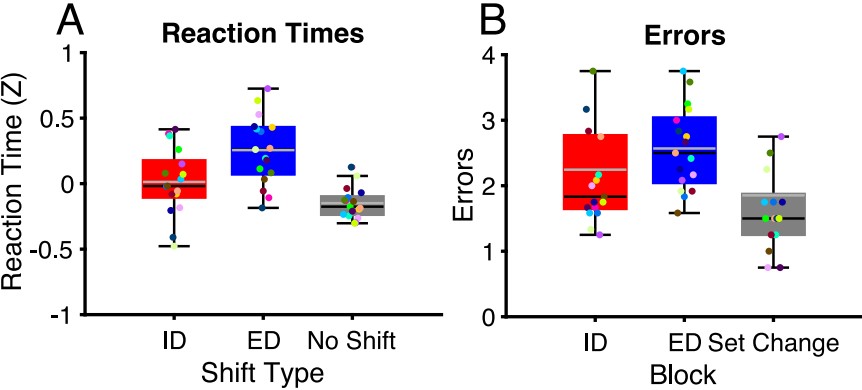

**Fig. 1 | Recording sites and EDID task. A** Positions of amygdala and hippocampus contacts that were used for analysis. Different colors represent different patients. **B** Trial procedure of EDID shifting task. On each trial, patients had to find which one of four stimuli, which varied on two dimensions (letters and shapes), was correct by pressing one of two buttons corresponding to its location on the screen and then monitoring the feedback. Once they had selected the correct stimulus for three trials in a row, it would then change either within (ID) or between (ED) stimulus dimensions. For example, it might change from T to S (ID) or T to ellipse (ED). After

six rule changes, a new stimulus set would appear. There were four stimulus sets in total. Two examples of the stimulus sets used are displayed in the top and bottom rows. Two choices were made on each trial so that the superimposition of the letters and shapes could be swapped to allow for calculation of which stimulus the patient believed to be the rule. The choice stimuli were presented for 2.1 s after which a response cue appeared signaling that they could respond. Correct and incorrect feedback was presented for .6 s. See methods for more details.

**Fig. 2 | Reaction times and errors in the EDID task. A** Boxplots showing mean and median (gray and black lines) reaction times across ED, ID and no shift conditions with range (whiskers) and inter-quartile range (box). The conditions are determined by patients' response on each trial relative to the previous trial. Different colored dots represent different patients (*N* = 17). **B** Boxplots showing mean and

median (gray and black lines) number of errors across ED, ID and set change conditions with range (whiskers) and inter-quartile range (box). Conditions are defined by the type of shift required to complete the block. ID Intradimensional, ED Extradimensional. Source data are provided as a Source Data file.

the fixed effects factor of block type (ED, ID, and stimulus set change) and random effects factor of patient. For errors, shift was based on block type rather than being determined by individual choices as errors are counted over multiple trials and defining shift based on individual trials could be confounded by differing numbers of shifts made irrespective of correctness. The Bonferroni adjusted *p*-value threshold of .0167 was used to correct for the three comparisons

made. One-tailed *p*-values were used based on the previous findings of increased errors on shift trials relative to set change trials and on ED shift trials relative to ID shift trials[5]. There were significantly more errors on ED shift trials relative to stimulus set change trials (t(1496) = −2.5, *p* = .0066, estimate = −.316, 95% CIs = [−.57 −.07]) (Fig. 2B). Although trending, there was no significant difference between ID shift and stimulus set change trials (t(1412) = −1.51, *p* = .07,

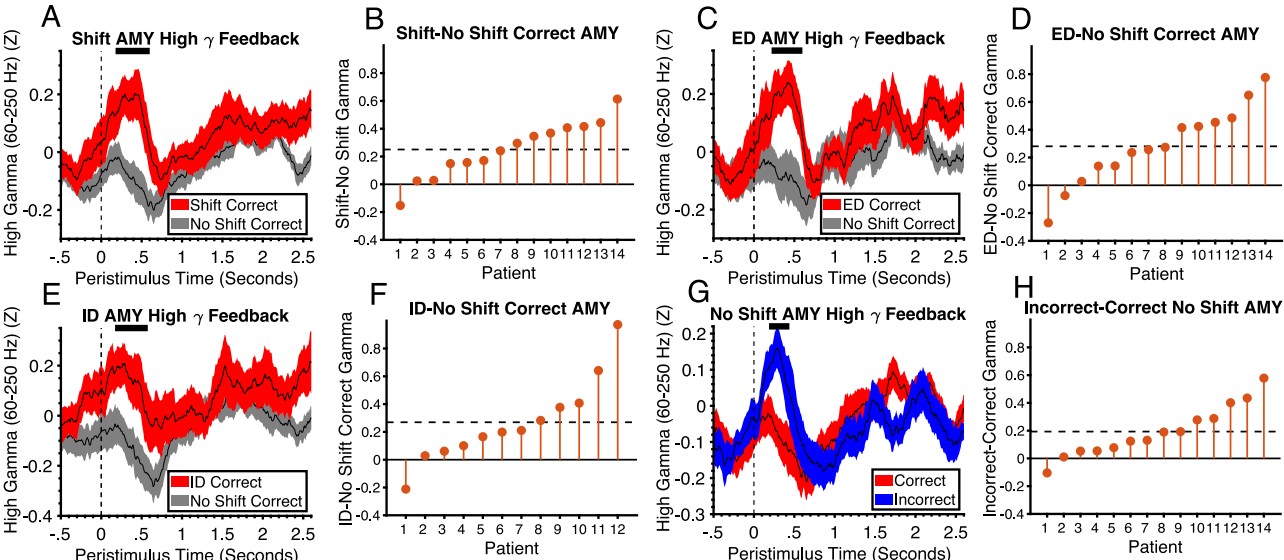

**Fig. 3 | Amygdala high gamma responses to feedback codes win-stay and lose-shift. A** High gamma activity in the amygdala in response to correct feedback on shift and no shift trials (black line represents the mean and shaded regions represent standard error and condition (see legend)). The horizontal black line at the top of the plot represents the time intervals of significant clusters (permutation test, $p = .0023$, FWEC, two-tailed). Vertical dashed line at t = 0 corresponds to feedback stimulus onset time. The feedback stimulus was presented for 600 ms after which was the inter-trial interval/ fixation. **B** Difference in mean activity between conditions across patients within significant time points shown in A ordered by size of effect. Horizontal dashed line is the mean difference across all patients. **C** High gamma activity in the amygdala in response to correct feedback on ED shift and no shift trials (black lines represent mean and shaded regions represents standard error and condition (see legend)). The horizontal black line at the top of the plot represents the time intervals of significant clusters (permutation test, $p = .0038$, FWEC, two-tailed). **D** Difference in mean activity between conditions across

patients within significant time points shown in (**C**). **E** High gamma activity in the amygdala in response to correct feedback on ID shift and no shift trials (black lines represent mean and shaded regions represents standard error and condition (see legend)). The horizontal black line at the top of the plot represents the time intervals of significant clusters (permutation test, $p = .0032$, FWEC, two-tailed). **F** Difference in mean activity between conditions across patients within significant time points shown in (**E**). **G** High gamma activity in the amygdala in response to correct and incorrect feedback on no shift trials (black lines represent mean and shaded regions represents standard error and condition (see legend)). The horizontal black line at the top of the plot represents the time intervals of significant clusters (permutation test, $p = .0049$, FWEC, two-tailed). **H** Difference in mean activity between conditions across patients within significant time points shown in (**G**). AMY Amygdala, ID Intradimensional, ED Extradimensional. Source data are provided as a Source Data file.

estimate = −.098, 95% CIs = [−.23 .029]) or ED and ID shift trials (t(2238) = 1.3, $p = .09$, estimate = .11, 95% CIs = [−.06 .28]). Consistent with previous studies[5], patients generally did not shift at rule change and almost always used the correct rule until receiving incorrect feedback (M = 92.7% of trials) as instructed. The number of correct trials which were not part of the three sequential correct trials required to reach criterion were very small (M = 2.4 trials across the whole task). These findings are all consistent with previous research demonstrating that patients are performing the task correctly and in a similar manner to healthy controls[5]. Next, we compare LFP activity and connectivity between conditions in the amygdala and hippocampus using t-tests and support vector machine (SVM) classification. P-values are corrected for multiple comparisons using cluster-based permutation tests unless otherwise specified.

## Feedback phase LFPs
We have previously shown HFG signals in the amygdala and hippocampus that code the immediate receipt of reward and punishment[28]. Therefore, we first analyzed the feedback phase of the task seeking to test if the HFG responses to correct and incorrect feedback are weighted differently at different transitions in the task to accurately deploy a win-stay-lose-shift strategy[33]. On shift trials, patients are searching for the correct stimulus, whereas on no shift trials they know which stimulus is correct. It is therefore hypothesized that correct feedback on shift trials would elicit a win-stay signal whereas on no shift trials nothing new needs to be learned and therefore the reward signal would be weaker or non-existent. This win-stay signal was evident in amygdala HFG activity. There was increased HFG activity to correct feedback on shift trials relative to no shift trials ($p = .0023$,

Cohens D = 1.24) (Fig. 3A, B). We next examined whether this difference was driven by ED and/ or ID shifts. For correct ID shifts, two patients were removed from the analysis due to low numbers of trials. Relative to no shift trials, there was a significant increase in HFG on correct ED shift trials ($p = .0038$, Cohens D = .998) (Fig. 3C, D) and correct ID shift trials ($p = .0032$, Cohens D = .88) (Fig. 3E, F). There was also larger activity on correct shift trials relative to incorrect shift trials (See Supplementary Fig. 2). However, this effect only approached significance ($p = .0375$) and did not differ for ED and ID shifts.

After patients have found the correct stimulus, they continue to choose this stimulus until they receive incorrect feedback. We hypothesized that incorrect feedback on no shift trials would elicit a lose-shift signal which indicates they should start shifting. This lose-shift signal was evident in amygdala HFG activity. On no shift trials, there was increased HFG amygdala activity in response to incorrect relative to correct feedback ($p = .0049$, Cohens D = 1.04) (Fig. 3G, H). This effect was not due to any perceptual differences between the correct and incorrect stimuli. The responses to correct and incorrect were in opposite directions across shift and no shift contexts and there was no significant difference between correct and incorrect across all types of shift trials. The increased activity to incorrect relative to correct was restricted to the no shift context as would be expected from a lose-shift signal. The finding of both win-stay and lose-shift signals in the amygdala is consistent with previous studies demonstrating its role in providing both reward and punishment reinforcement learning signals[8,28,34]. However, the responses to reward and punishment were dependent on whether the patient had made a shift. This suggests that patients may implement a strategy, where they are more sensitive to reward after a shift and punishment after a stay. Staying and shifting

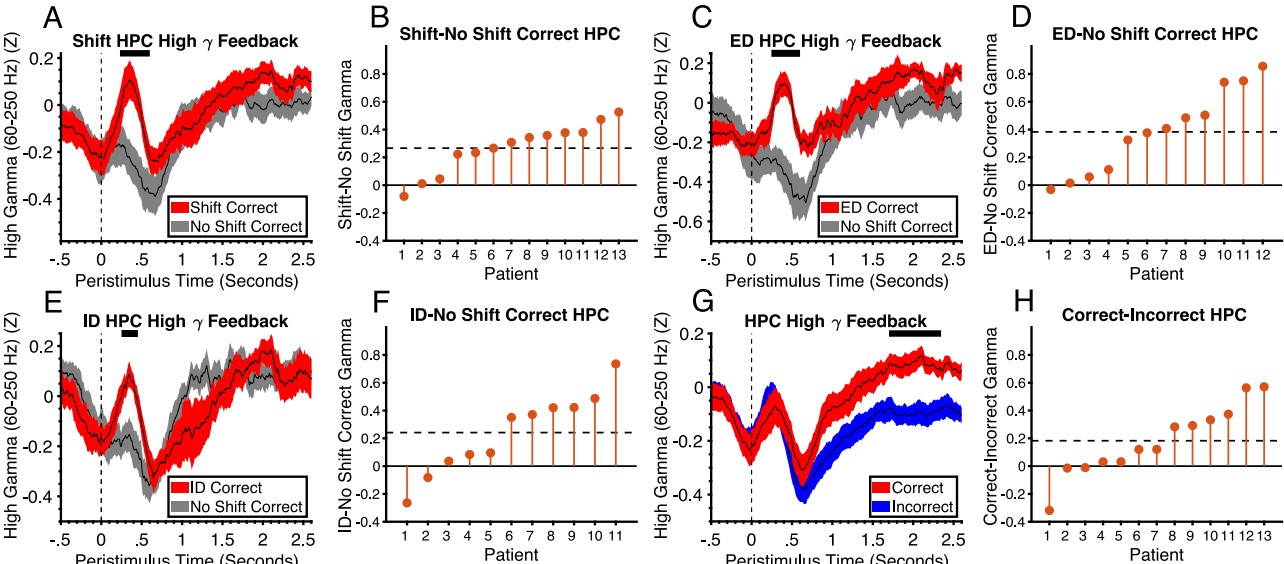

**Fig. 4 | Hippocampal high gamma responses to feedback codes win-stay. A** High gamma activity in the hippocampus in response to correct feedback on shift and no shift trials (shaded regions represent standard error and condition (see legend)). The horizontal black line at the top of the plot represents the time intervals of significant clusters (permutation test, $p = .0017$, FWEC, two-tailed). **B** Difference in mean activity between conditions across patients within significant time points shown in (**A**). **C** High gamma activity in the hippocampus in response to correct feedback on ED shift and no shift trials (black lines represent mean and shaded regions represents standard error and condition (see legend)). The horizontal black line at the top of the plot represents the time intervals of significant clusters (permutation test, $p = .0027$, FWEC, two-tailed). **D** Difference in mean activity between conditions across patients within significant time points shown in (**C**). **E** High gamma activity in the hippocampus in response to correct feedback on ID shift and no shift trials (black lines represent mean and shaded regions represents standard error and condition (see legend)). The horizontal black line at the top of the plot represents the time intervals of significant clusters (permutation test, $p = .023$, FWEC, two-tailed). **F** Difference in mean activity between conditions across patients within significant time points shown in (**E**). **G** High gamma activity in the hippocampus on correct and incorrect feedback trials showing effect in the ITI (black lines represent mean and shaded regions represents standard error and condition (see legend)). The horizontal black line at the top of the plot represents the time intervals of significant clusters (permutation test, $p = .017$, FWEC, two-tailed). **H** Difference in mean activity between conditions across patients within significant time points shown in (**G**). HPC hippocampus, ID Intradimensional, ED Extradimensional. Source data are provided as a Source Data file.

are beneficial to patients when they know and do not know the rule which in turn can only be known from reward and punishment. On stay trials, patients already know the rule and so responses to reward confer no further benefits whereas punishment signals that they should start shifting. In contrast, on shift trials, patients do not know the rule and so reward is more important than when they are not shifting and know the rule. One means of implementing such a strategy is to pay more attention to reward in the shifting phase and more attention to punishment in the staying phase. The amygdala may be the optimal region to implement this type of attentional selection. It receives highly processed visual input from the temporal lobes and contains representations of both the reward and punishment value of stimuli. It may filter incoming visual information for both the visual properties of the feedback stimulus and its value depending on shift context. This is highly consistent with previous studies demonstrating interactions between value coding and attention in the amygdala[12–14].

In the hippocampus, there was a significant decrease in HFG around the time of feedback across all trials relative to baseline (FDR corrected $P < .025$, One-sample Wilcoxon signed ranks test, Cohens D = 1.54) (Supplementary Fig. 3A, B). However, the win-stay signal seen in the amygdala was also seen in the hippocampus. There was a significant increase in HFG on correct shift relative to correct no shift trials ($p = .0017$, Cohens D = 1.5) (Fig. 4A, B) and on correct compared to incorrect shift trials ($p = .0061$, Cohens D = 1.07) (Supplementary Fig. 3C, D). We then examined whether these effects were driven by ED or ID shifts. Due to low trial numbers, two patients were removed from the analysis of correct ID shifts and one patient was removed from analysis of correct ED shifts. There was increased HFG activity on correct ED shift trials relative to correct no shift trials ($p = .0027$, Cohens D = 1.27) (Fig. 4C, D) and on correct relative to incorrect ED shift trials ($p < .001$, Cohens D = 1.81) (Supplementary Fig. 3E, F). There

was also increased HFG activity on correct ID shift trials relative to correct no shift trials ($p = .023$, Cohens D = .83) (Fig. 4E, F) and on correct relative to incorrect ID shift trials ($p = .012$, Cohens D = 1.04) (Supplementary Fig. 3G, H). The pattern of responses in the hippocampus is consistent with a role in using reward to update the correct rule in a cognitive map of task space. Unlike the amygdala, there was no lose-shift signal in the hippocampus, consistent with previous studies showing selective hippocampal HFG responses to reward and not punishment and suggesting involvement in using reward to update rules in memory rather than reinforcement value representation[28,35]. This was further corroborated by comparing the lose shift signal between amygdala and hippocampus. We subtracted activity between incorrect and correct no shift trials, averaged over the time window of the effect and compared between the amygdala and hippocampus using a non-parametric Wilcoxon rank sum test. A between subjects test was used due to some patients having amygdala but not hippocampus electrodes and vice versa and a non-parametric test was used due to potential non-normality of HFG. The difference between incorrect and correct on no shift trials was significantly larger in the amygdala compared to the hippocampus ($p = .022$, one-tailed, Cohens D = .61). The response to correct feedback was dependent on shift context. There was no significant difference between correct and incorrect feedback across both shift and no shift trials (Fig. 4G). However, there was a significant increase in HFG on correct relative to incorrect trials in the subsequent inter-trial-interval (ITI) ($p = .017$, Cohens D = .73) (Fig. 4G, H). In this task, it is crucial for patients to maintain a memory trace of the rule across trials. Therefore, this effect may reflect maintenance, rehearsal, or consolidation of the correct rule in memory across trials. Similar rule memory signals in the ITI have been demonstrated in neuronal recordings from DLPFC in monkeys[3]. Interestingly, this effect ramped up gradually beginning at the time of

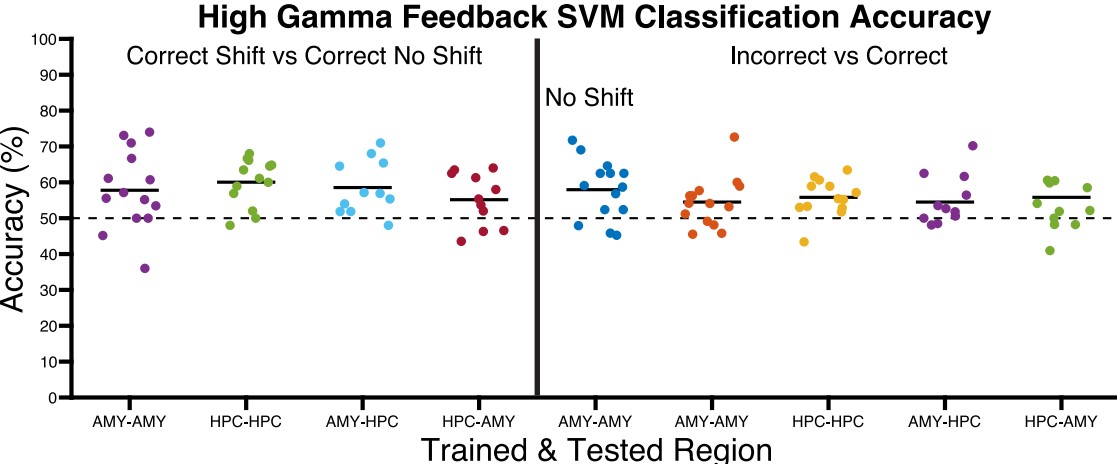

**Fig. 5 | Multivariate patterns of high gamma activity in response to feedback can be used to predict condition membership.** Scatter graphs showing predictive accuracy of support vector machines' (SVMs) ability to differentiate feedback conditions when trained and tested on amygdala and hippocampus. The dots on the y axis represent the performance of individual patients' SVM's. The x axis shows the train and test region combination. The left panel shows the accuracy of SVM's when classifying correct shift vs correct no shift conditions. The right panel shows the accuracy of SVM's when classifying correct vs incorrect conditions on no shift trials specifically and across all incorrect and correct trials. Horizontal lines indicate mean classification accuracy. Fourteen patients were trained and tested using the amygdala, thirteen patients were trained and tested using the hippocampus and eleven patients were tested using amygdala and trained using hippocampus and vice versa. AMY Amygdala, HPC Hippocampus. Source data are provided as a Source Data file.

feedback reception (See Supplementary Fig. 4) perhaps because a stronger memory trace was needed as time passed to prevent decay or due to increased preparation as the next trial became more imminent. In summary, the findings are consistent with the idea that the amygdala filters the correct rule into the hippocampal memory system by attending to reward.

## Support vector machine classification analysis

In addition to mass univariate analyses, we also performed complementary multivariate analyses using a binary support vector machine (SVM) classifier. This analysis is qualitatively distinct from previous analyses in that it seeks to test whether we could predict condition membership trial-by-trial within and across regions using patterns of HFG across all time points simultaneously at the expense of not being able to pinpoint exactly which time points contributed to significance. By training on individual regions we were able to test if information distributed throughout the duration of trials could be used to distinguish conditions, thereby demonstrating its importance in coding the processes important for those conditions. By training the SVM within one region and testing it on another, we were able to test similarity and interdependence in the information encoded in both regions and therefore potential cooperation in such processes. This was performed on 11 patients who had both amygdala and hippocampus electrodes. Performance of the classifier within each region was cross-validated using a leave-one-out design. Percentage correct classification was calculated for each patient and tested for significance using a one-sample, Wilcoxon signed ranks test against chance (50%). One-tailed p-values were used as significant accuracy would be above chance only and were corrected for multiple comparisons using the Bonferroni-Holm correction. We repeated the same contrasts performed with the mass univariate analyses with SVM. As the feedback phase was very brief, we included in the analysis both the feedback phase and subsequent ITI (i.e. the time-series depicted in Figs. 3 & 4), as feedback activity may be sustained into the ITI and contain useful information patterns that could differentiate conditions. We first tested whether we could use feedback activity to predict correct shift and no shift conditions as we found similar patterns of activity across ED and ID conditions in previous analyses and combining the two types of shift trials provided more power for this

analysis. We were able to significantly predict condition membership within the amygdala (Mean accuracy=57.4%, range = [36 74], SD = 13.1%, $p$ = .017, Cohens D = .71) and hippocampus (M = 60.2%, range = [48 68], SD = 7.3%, $p$ = .001, Cohens D = 1.5) using HFG activity (Fig. 5). We next tested if we could use amygdala activity to classify hippocampal activity and vice-versa. Indeed, after training the classifier on amygdala activity, we were able to significantly predict which condition hippocampal activity belonged to (M = 58.5%, range = [48 71], SD = 7.5%, $p$ = .0024, Cohens D = 1.1) and vice versa (M = 55.2%, range= [43.6 64], SD = 7.4%, $p$ = .029, Cohens D = .7). There was also significant classification of correct and incorrect conditions on no shift trials in the amygdala (M = 58%, range = [45 71.7], SD = 8.3%, $p$ = .0025, Cohens D = .96), but not in the hippocampus or across regions (all P's >.05). There was also significant classification of correct and incorrect trials regardless of shift within the amygdala (M = 54.5%, range = [45.5 72.6], SD = 7.7%, $p$ = .017, Cohens D = .64) and hippocampus (M = 55.8%, range = [43.4 63.4], SD = 5.2%, $p$ = .0023, Cohens D = 1.1) and across regions when training on amygdala and testing on hippocampus (M = 55.1%, range = [48 70.2], SD = 7.5%, $p$ = .014, Cohens D = .73) and vice versa (M = 53.2%, range = [41 60.6], SD = 6.3%, $p$ = .042, Cohens D = .51). In summary, these findings converge with those of the mass univariate analyses, showing that win-stay can be predicted within and across regions whereas lose-shift was restricted to the amygdala. To a lesser extent, we were also able to differentiate correct from incorrect conditions across all trials within and across regions. This was expected from the hippocampus, which showed differences in the ITI in the mass univariate analyses. The finding in the amygdala and across regions may be due to the ability of SVM to make use of patterns of activity that do not reach significance individually but nevertheless contain useful information when combined.

## Synchrony between amygdala and hippocampus

The similarity of the win-stay HFG response in the amygdala and hippocampus and the intra- and cross-region SVM classifications suggest that the two regions are coupled into a functional network. One means by which this could be achieved is through low frequency phase coherence which may synchronize processing in each region in time[36–38]. To test this hypothesis, we examined phase-locking between the two regions which measures the consistency in the phase

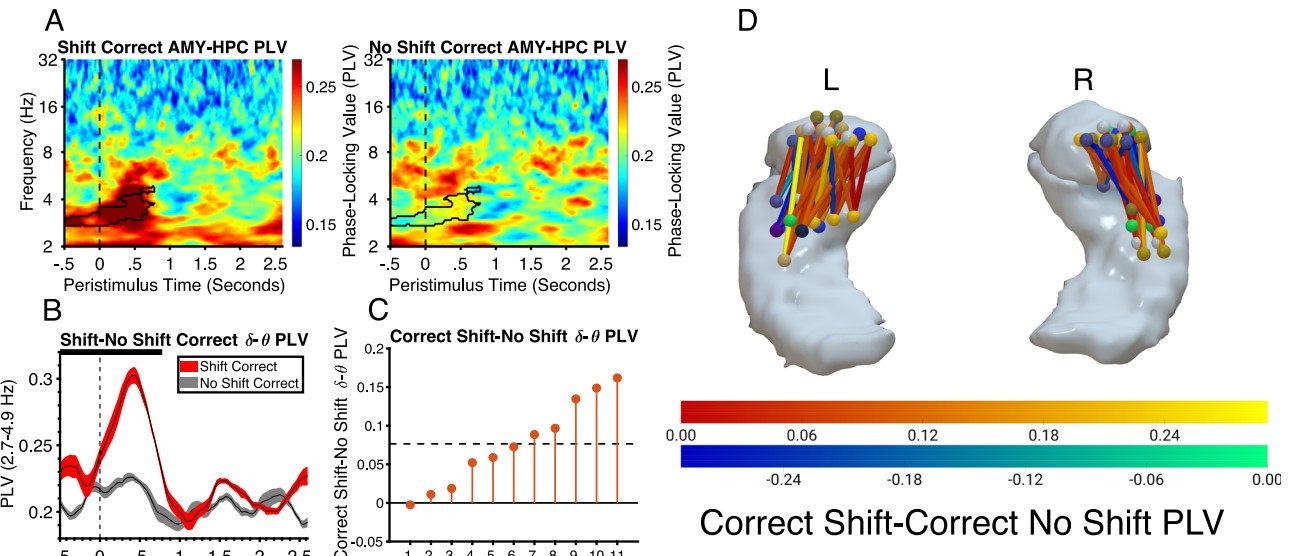

**Fig. 6 | Amygdala and hippocampus activity is synchronized during win-stay via phase-locking in the delta-theta band. A** Time-frequency plots, showing phase-locking values (PLV) between amygdala and hippocampus across correct trials in the shift and no shift conditions. Black outline highlights the significant cluster (permutation test, p = .0015, FWEC, two-tailed). Vertical dashed line at t = 0 corresponds to feedback stimulus onset time. **B** PLV between amygdala and hippocampus for correct feedback on shift and no shift trials averaged over the frequency range spanned by the significant cluster shown in A (shaded regions represent standard error and condition (see legend)). The horizontal black line at the top of the plot represents the time intervals of significant clusters. **C** Difference in mean activity between conditions across patients within significant time points shown in **A** and **B** ordered by size of effect. Horizontal dashed line represents mean difference. **D** Network of phase-locking between each pair of channels for the contrast of correct shift vs correct no shift overlaid on the amygdala and hippocampus. The color of the connections represents the magnitude of the difference in phase-locking between correct shift and correct no shift conditions in the significant time window highlighted in panels **A** and **B**. AMY Amygdala, HPC Hippocampus, PLV Phase-locking values. Source data are provided as a Source Data file.

differences between oscillations in each region across trials[39]. As low frequency oscillations tend to be band-specific (occurring in delta (1–4 hz), theta (4–8 hz), alpha (8–12 hz) and beta (12–30 hz)) we performed time-frequency analysis of PLV between 2 and 32 Hz to identify the specific frequency that showed synchrony. This was performed on the 11 patients who had both amygdala and hippocampal electrodes. There was significantly increased phase-locking in the delta-theta range (2.7–4.9 Hz, $p < .001$, Cohens D = 1.4) between amygdala and hippocampus in the correct shift condition relative to the correct no shift condition. This effect occurred at approximately the same time as differences were observed in the HFG range for each region separately (Fig. 6). We have previously demonstrated delta phase-locking between amygdala and hippocampus during receipt of reward[28]. However, the findings from the current study suggest that this coupling can be dependent on the goals and attentional set of the patient in a similar way to the HFG responses in each region individually described above.

The feedback phase was preceded by a button press response. However, this occurred on all trials and therefore is equivalent across conditions. In order to be certain that the detected effects were not driven by response, we performed the same analyses on the first button press which was not followed by feedback. There were no significant differences between conditions showing that the responses were not driving effects of feedback.

**Decision phase LFPs**

We next analyzed the decision phase. In the hippocampus, HFG activity across all trials was generally decreased relative to baseline like in the feedback phase ($P < .025$, FDR corrected, One-sample Wilcoxon signed ranks test, Cohens D = 1.29) (Fig. 7A, B). This was not the case in the amygdala and a direct comparison of amygdala with hippocampus using an independent samples t-test with 5000 permutations showed greater suppression in the hippocampus ($p = .015$, Cohens D = .99) (See Supplementary Fig. 5). In the hippocampus, this decrease was greater

on no shift trials relative to shift trials ($p = .0035$, Cohens D = 1.1) (Fig. 7C, D). The difference was driven by both ID ($p = .0095$, Cohens D = .86) (Fig. 7E, F) and ED shifts ($p = .015$, Cohens D = .99) (Fig. 7G, H). The patterns of activity in the hippocampus are consistent with a role in forming, updating, and maintaining the rule. The inhibitory activity may be involved in preventing interference between competing representations and the rule. On shift trials in the decision phase, this inhibitory activity is less because patients are changing their hypothesis about which stimulus is correct which requires increased competition between representations[40]. At the time of correct feedback, the model can be changed and the rule committed to and therefore there is a release from inhibition. The inhibitory HFG responses in the hippocampus may be necessary to hold a cognitive map of task rules in working memory without interference from incoming signals from the senses or irrelevant internal representations in memory. This idea is also consistent with the known role of the hippocampus in pattern separation[41]. In this case it is involved in separating the rule representation from non-rule representations by inhibiting the latter.

We also examined differences between conditions in low-frequency oscillatory power which is known to play an important role in cognitive processing[42–44]. To do this, we performed time-frequency analysis on low frequency power. There was a significant increase in theta (3.4–6.1 Hz) activity on no shift trials relative to shift trials in the amygdala ($P < .001$, Cohens D = 1.6) (Fig. 8A–C). This effect was seen on both ED ($p < .001$, Cohens D = 1.3) and ID ($P < .025$, Cohens D = 1.03) shift trials (See Supplementary Fig. 6) which were not significantly different from each other. The effect appeared to be sustained throughout the trial as it was also significant in the outcome phase and subsequent ITI ($p < .0001$, Cohens D = 1.6) (See Supplementary Fig. 7). A similar effect was seen in the hippocampus in the delta band ($p = .014$) (Fig. 8D–F). However, this difference was driven primarily by a decrease to ED relative to no shift ($p < .001$) and not ID relative to no shift, although there was no significant difference between ED and ID (See supplementary Fig. 8). Like the amygdala, this

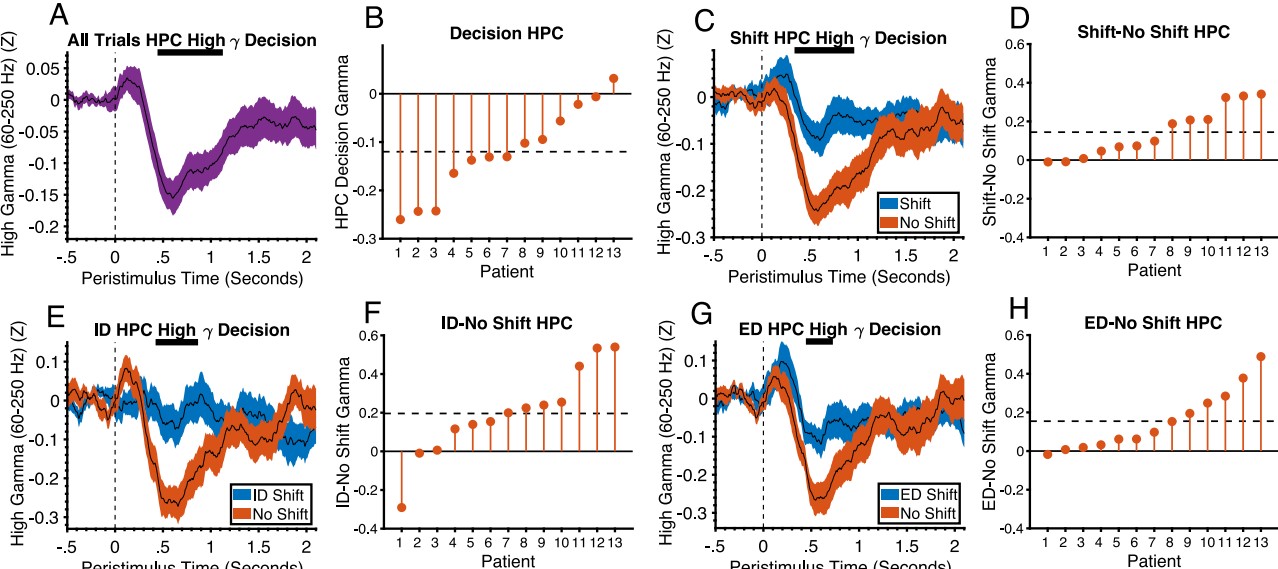

**Fig. 7 | Hippocampal high gamma activity in the decision phase is inhibited to prevent interference in rule representation. A** High gamma activity in the hippocampus during the decision phase relative to baseline across all trials (black line represents mean and shaded region represents standard error. The horizontal black line at the top of the plot represents the time intervals of significant differences from baseline (Wilcoxon's signed ranks test, $p < .025$, FDR corrected, two-tailed). Vertical dashed line at t = 0 corresponds to choice stimulus onset time. The choice phase was 2.1 s. **B** Mean activity across patients within significant time points shown in A ordered by size of effect. Horizontal dashed line is the mean difference across all patients. **C** High gamma activity in the hippocampus during the decision phase of shift and no shift trials (black lines represent mean and shaded regions represents standard error and condition (see legend)). The horizontal black line at the top of the plot represents the time intervals of significant clusters (permutation test, $p = .0035$, FWEC, two-tailed). **D** Difference in mean activity between conditions

across patients within significant time points shown in (**C**). **E** High gamma activity in the hippocampus during the decision phase of ID shift and no shift trials (black lines represent mean and shaded regions represents standard error and condition (see legend)). The horizontal black line at the top of the plot represents the time intervals of significant clusters (permutation test, $p = .0095$, FWEC, two-tailed). **F.** Difference in mean activity between conditions across patients within significant time points shown in (**E**). **G** High gamma activity in the hippocampus during the decision phase of ED shift and no shift trials (black lines represent mean and shaded regions represents standard error and condition (see legend)). The horizontal black line at the top of the plot represents the time intervals of significant clusters (permutation test, $p < .015$, FWEC, two-tailed). **H** Difference in mean activity between conditions across patients within significant time points shown in (**G**). HPC hippocampus, ID Intradimensional, ED Extradimensional. Source data are provided as a Source Data file.

effect was sustained into the outcome phase (See supplementary Fig. 9). There was no significant difference between amygdala and hippocampus in this effect. Therefore, when patients know the rule, there is a shift from high gamma responsivity to reward to theta/ delta oscillations. It is interesting to note that the no shift phase of the block is a relatively long period of time relative to the individual events and that this is tracked by low frequency oscillations which also occur on a much longer time scale. Such slower oscillations may reflect the maintenance of the correct rule in working memory over several trials[45,46]. This working memory representation may be necessary for the decision-making process which has also been shown to involve the amygdala[10]. The long duration effect is also consistent with the theory that the amygdala is involved in coding reward contingencies over long time periods[9]. It is also noteworthy that the difference in delta/ theta was found for the same contrast as the suppression of HFG in the hippocampus which may be attributable to pattern separation. This is because previous studies have also highlighted a role for amygdala and hippocampus delta/ theta activity in pattern separation of emotional memories[44].

In primates, amygdala neurons are also known to respond to novel stimuli[9]. Our task allowed us to test for similar responses in humans. There were four unique stimulus sets that were initially unfamiliar to the patients which allowed us to compare amygdala responses in the first block of trials, when the stimuli were novel, with the last block, when they were familiar. There was increased amygdala HFG activity in response to the choice stimuli on trials in the first block after stimulus set change (early trials) relative to the last block (late trials) ($p = .0024$, Cohens D = 1.3) (Fig. 9A, B). The amygdala responses to novelty may reflect the formation of stimulus memories,

particularly in the basolateral region, which may function to facilitate future associations with reward or punishment. It may also reflect a reinforcing signal that motivates their exploration and discovery of potentially rewarding properties. Although the effect of novelty did not reach significance in the hippocampus, it was trending ($p = .07$) and was not significantly different from the amygdala. The trending effect showed the same peak timing as the amygdala but was weaker and superimposed on top of the general decrease seen across all trials (see Supplementary Fig. 10). The finding of decision-making and novelty signals, in addition to signals related to value coding, reinforcement learning and attention, is consistent with the multidimensional nature of responses seen in the primate amygdala[15,16].

We did not find any significant SVM classifications in the decision phase (all p's > .05). This may be because reward and punishment responses are evoked by the stimulus in a more exogenous fashion and are therefore more consistent across trials whereas decision-related activity is more endogenously generated and may vary more.

## Discussion

In this study, we investigate the brain systems that form and use different types of rules using an ED and ID shifting task. Consistent with previous studies, reaction times were slower on ED trials relative to ID trials which were in turn slower than no shift trials and errors were greater on ED shift trials relative to set change trials[5]. There were also trends for greater errors on ID shift trials relative to set change trials. Because ED, ID and set change trials can logically be solved within the same number of trials, any differences between them must be due to the tendency to perseverate to the previous target or the previous dimension. Adaptive behavior depends on the

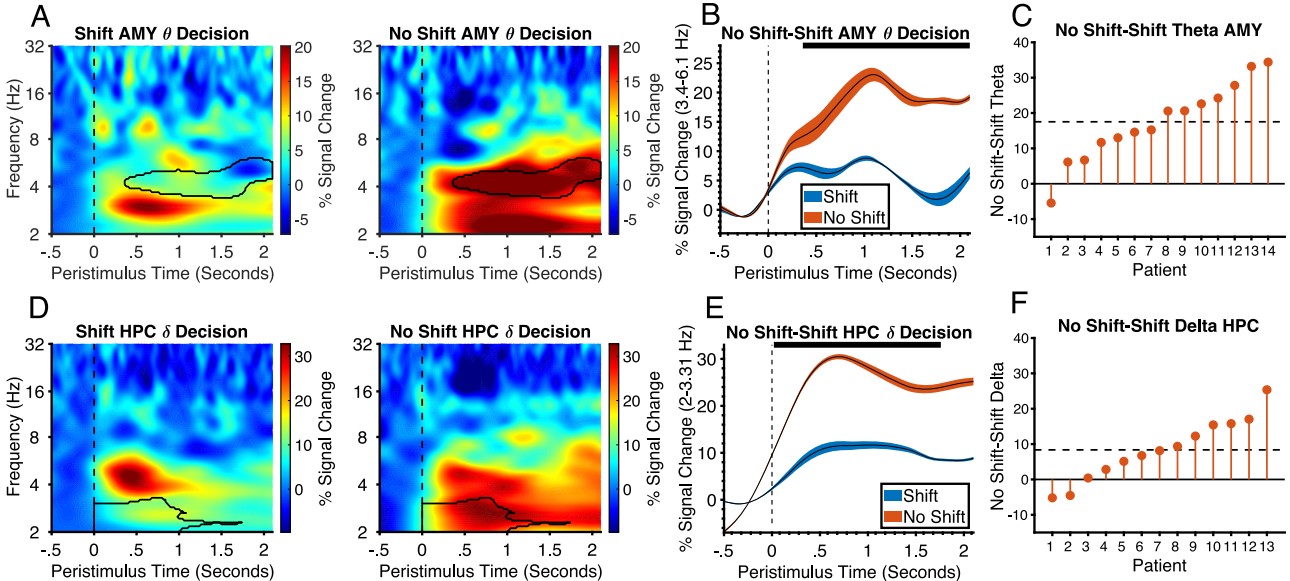

**Fig. 8 | Low frequency power in amygdala and hippocampus is involved in decision-making when the rule is known. A** Time-frequency plots showing differences in amygdala theta power in the decision phase between shift and no shift conditions. Black outline highlights the significant cluster (permutation test, $p < .001$, FWEC, two-tailed). Vertical dashed line at $t = 0$ corresponds to choice stimulus onset time. The decision phase was 2.1 s. **B** Theta activity in the amygdala in the decision phase on shift and no shift trials averaged over the frequency range spanned by the significant cluster shown in **A** (black lines represent mean and shaded regions represents standard error and condition (see legend)). The horizontal black line at the top of the plot represents the time intervals of significant clusters. **C** Difference in mean activity between conditions across patients within significant time points shown in A and B ordered by size of effect. Horizontal dashed line is the mean difference across all patients. **D** Time-frequency plots showing differences in hippocampal delta power in the decision phase between shift and no shift conditions. Black outline highlights the significant cluster (permutation test, $p = .014$, FWEC, two-tailed). **E** Delta activity in the hippocampus in the decision phase on shift and no shift trials averaged over the frequency range spanned by the significant cluster shown in D (black lines represent mean and shaded regions represents standard error and condition (see legend)). **F** Difference in mean activity between conditions across patients within significant time points shown in D and E ordered by size of effect. Horizontal dashed line is the mean difference across all patients. AMY Amygdala, HPC hippocampus, ID Intradimensional, ED Extradimensional. Source data are provided as a Source Data file.

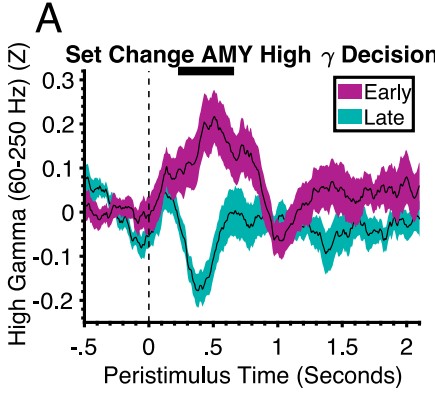
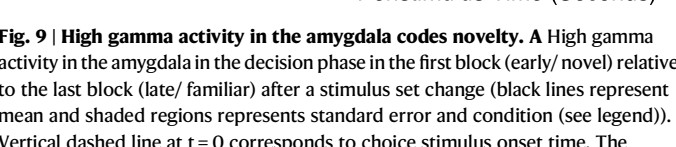
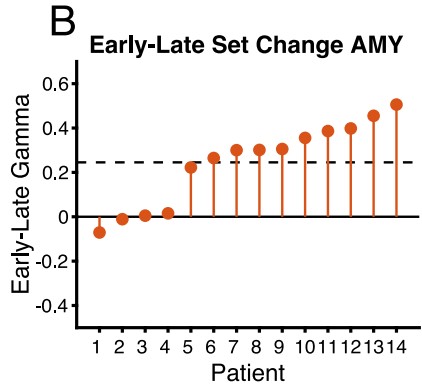

**Fig. 9 | High gamma activity in the amygdala codes novelty. A** High gamma activity in the amygdala in the decision phase in the first block (early/ novel) relative to the last block (late/ familiar) after a stimulus set change (black lines represent mean and shaded regions represents standard error and condition (see legend)). Vertical dashed line at $t = 0$ corresponds to choice stimulus onset time. The horizontal black line at the top of the plot represents the time intervals of significant clusters (permutation test, $p = .0024$, FWEC, two-tailed). **B** Mean difference in activity between conditions across patients within significant time points shown in A ordered by size of effect. Horizontal dashed line is the mean difference across all patients. AMY Amygdala. Source data are provided as a Source Data file.

ability to over-ride these tendencies by performing an ED or an ID shift, processes which may be impaired in certain neuropsychiatric disorders[1,2]. It is therefore crucial to understand the brain systems involved in these processes. Most research has focused on the contribution of the PFC to these processes due to its role in reversal learning and executive function[3–5]. However, rule use may be broken down into a range of sub-processes which may be coded in brain regions such as amygdala and hippocampus which neurophysiological recording studies in primates have shown to be involved in multi-dimensional coding[15,16] and cognitive maps of task space[17,18].

Therefore, we examined the role of the human amygdala and hippocampus in ED and ID rule shifting using intracranial recordings.

Our findings suggest an alternative way of thinking about how shifting is accomplished by the brain (Fig. 10). When patients are making choices, they may form hypotheses about which stimulus is correct and then commit to the rule shift at the point it is confirmed with correct feedback. The hippocampus appears to be key to this process. On no shift trials, it was inhibited, perhaps to prevent interference between the current task set and competing representations[40]. This could be similar to inhibitory tagging of irrelevant items in visual

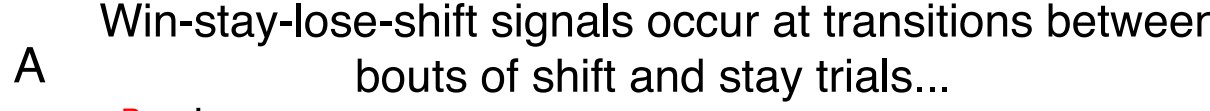

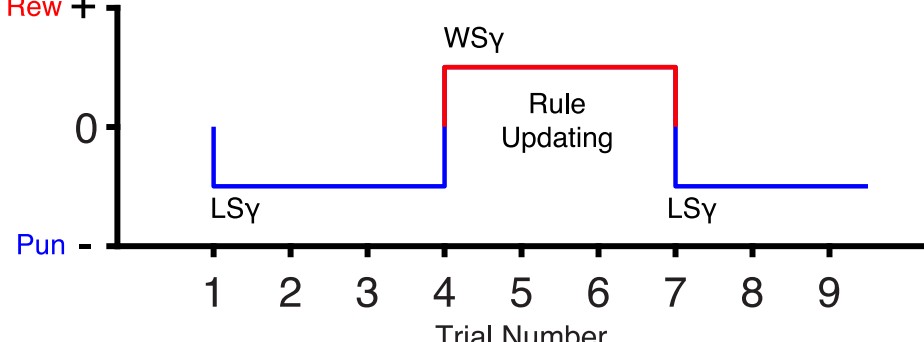

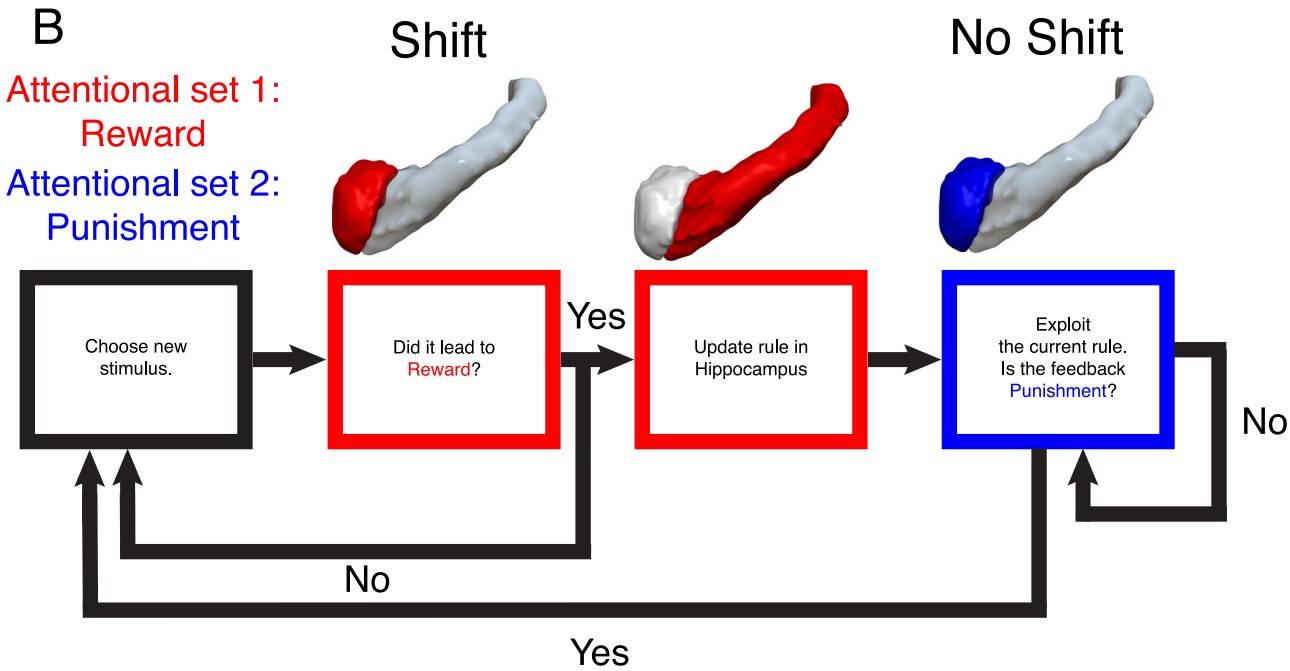

**Fig. 10 | Summary of shifting processes in the amygdala and hippocampus.**
**A** Our results suggest HFG win-stay-lose-shift signals occur at transitions between bouts of shifting and staying trials and are controlled by attention. This may have the advantage of being more economical than responding on every trial thereby conserving cognitive and neural resources as the rule needs to be updated less often. Additionally, within bouts of stay trials, delta-theta oscillations in the amygdala and hippocampus are involved in maintaining a signal that the current rule is rewarded and can be used to guide decisions and hippocampus HFG is inhibited to prevent interference in rule representation. **B** Participants are proposed to adopt two different strategies and attentional sets at different phases of the task as shown by patterns of HFG activity. In the search phase of the task, the amygdala is tuned to attend more to reward feedback (red), at least relative to the staying phase of the task. If reward feedback is received, the correct rule is updated in the hippocampus and patients move to the second strategy of choosing the same stimulus while the amygdala attends more to punishment (blue) relative to reward. Once punishment is received patients change back to the first strategy. Rew Reward, Pun Punishment, WS Win-Stay, LS Lose-Shift.

search[47] or the known pattern separation function of the hippocampus which may separate the rule from non-rule stimuli which may have similar perceptual features (intra-dimensionally) or be overlapping in space (extra-dimensionally)[41]. However, on shift trials, the hippocampus was activated by reward (win-stay), suggesting confirmation of the hypothesis and plastic changes to the internal model. In addition, there was a weakening of inhibition during choice on shift trials which may allow for greater flexibility when making hypotheses. The amygdala showed a similar win-stay response as the hippocampus, but it also showed a lose-shift signal consistent with a role in both reward and punishment coding and learning which may be used by the hippocampus for its functions[8,28,34,48].

The reinforcement signals must be selected by attention as they were only seen at transitions in the task that would be expected if subjects were using a win-stay-lose-shift strategy. On shift trials, patients do not know the rule and are exploring until correct feedback signals they should stay with this rule. In contrast, on no shift trials, patients know the correct rule and are staying until incorrect feedback

signals they should start exploring for the new rule. Therefore, it makes sense that patients are more sensitive to reward in the former situation and punishment in the latter. The strategy of only responding at critical transitions in the task is highly economical and may conserve cognitive and neural resources. This is consistent with the idea that goals may bi-directionally interact with attention and emotion. Emotions may set the goals for behavior and goals may determine what we attend and emote to. This has been demonstrated in the field of visual attentional control settings which has demonstrated that attentional capture of a stimulus depends on its goal relevance[49–51]. Indeed, cells in the primate amygdala are often selective for both high value and presentation in contralateral visual space and their firing correlates with behavioral measures of attentional orienting[12–14]. Our findings also demonstrate reward and punishment signals in human amygdala and hippocampus can be determined by goals and attentional set to enable accurate credit assignment. The attentional sets for correct and incorrect feedback processing may in turn be determined by task sets which may differ between the switching and staying phases and be configured through to*p*-down influences from the PFC[43]. Given the amygdala is also crucially involved in stimulus-reinforcer devaluation, it is also possible that the decreased sensitivity to reward on no shift trials reflects devaluation of the feedback[52]. The attention and devaluation accounts may not be mutually exclusive. It may be the attentional modulation of the amygdala response that increases or decreases the valuation. We also note that these effects may be related to the deterministic nature of our task. Patients could have learned that once they have found the rule it will be correct for another two trials. This knowledge could act as a to*p*-down attentional influence and be further evidence of patients' ability to form cognitive maps of task space and the role of amygdala and hippocampus in these processes. In a probabilistic situation, the reward signals may behave differently.

The increased hippocampal activity for win-stay, is consistent with the idea that the hippocampus maintains cognitive maps of task space[17–19]. An important question about shifting is when after reinforcement change does a shift occur? Although our task has predefined decision-making and feedback phases, the way the brain shifts may not conform to our pre-existing expectations. It may not be useful to think of shifting as an individual process that occurs in the decision-phase. Instead, it may be better to break the process down into sub-processes that are distributed throughout the trial. In the decision phase, patients may well choose a stimulus that conforms to an ED or an ID shift, but it does not make sense for the patient to commit to the rule shift until they know for certain that it is correct. The choice may be more akin to a hypothesis test and the rule shift may actually occur at the point of correct feedback. However, unlike the amygdala, the hippocampus did not show a lose-shift signal. This highlights an important distinction between the two regions. Whereas the hippocampus may be involved in updating the rule in memory when it is known, as signaled by correct feedback, the amygdala is more involved in coding both positive and negative outcome values. The process of generating a hypothesis may be akin to exploration and require initiation by the medial PFC[53] followed by visual attentional shifts and working memory maintenance by the VLPFC[3–5]. However, it's representation as a rule may be weak until it is confirmed to be correct.

Whilst our HFG findings are consistent with the idea of a dissociation between amygdala and hippocampus in terms of reinforcement learning and cognitive map representation, our findings also suggest the two processes are integrated via delta-theta phase-locking. This may allow each regions individual processes, reflected in HFG activity, to be synchronized in time and in sequence and for reinforcement value to be transferred from the amygdala to the hippocampus and for information about task space to be transferred from hippocampus to amygdala[38]. This finding is also supported by our machine learning analyses which allowed us to distinguish correct shift trials from no shift trials in the hippocampus using amygdala activity and vice versa, again suggesting coupling and information transfer. We have previously demonstrated amygdala and hippocampal HFG increases and delta phase-locking between the two regions in response to reward receipt[28]. The current findings suggest that this is not obligatory and determined by the patients' goals and attentional set.

The finding of shifting signals in the hippocampus is consistent with available evidence. A lesion study in humans showed that although ED shifting was not impaired after unilateral amygdalo-hippocampectomy, reaction times were slower on ED trials and ED shifting could still be performed by the remaining hemisphere[54]. Monkeys with bilateral hippocampal, but not amygdala, lesions were impaired in learning new abstract rules but not in using rules formed before ablation[55]. The lack of effect for the amygdala does not mean it is not involved in this process as subjects can compensate by using visually based performance rules[56]. While using visual instead of reward and punishment information may be sufficient to perform simple cognitive tasks, it may not be as effective in social contexts which are more complex and nuanced. The finding that the amygdala was involved in learning from reward after an ED shift as well as an ID shift is consistent with primate work. The amygdala has been shown to use abstract representations like that required to evaluate an ED shift. When reward associations are reversed between two contexts, primates can use context to adjust behavior more rapidly than they would if they were only learning from outcomes. Amygdala activity coded context and the context signal of one conditioned stimulus could be predicted from the other suggesting that they are linked together in a set[11].

Hippocampal HFG activity was also larger in the ITI after receiving correct relative to incorrect feedback. To perform the task correctly, patients must get 3 trials correct in a row. This requires that the patient remembers the correct rule in between trials. We believe that the increased activity in the ITI to correct feedback reflects maintenance, rehearsal or consolidation of the rule in memory so that it is not forgotten in between trials. Similar types of rule working memory signals have been demonstrated in the ITI in the DLPFC in macaques during the WCST[3]. Sustained HFG signals in the OFC have also been shown to carry reward-related information from previous trials into future trials to influence behavior[27] and the OFC is also known to be involved in representing cognitive maps of task space, principally through its connectivity with the hippocampus[17,18,57]. The effect in the ITI emerged gradually after feedback which is consistent with studies demonstrating ramping dynamics in high gamma activity[58,59].

Several other types of signals were also found. Delta/theta activity was larger in the amygdala and hippocampus throughout no shift trials relative to shift trials which may also reflect working memory for the correct rule[43,45,46] in service of decision-making[10]. It would be interesting to know what type of representation is being maintained. It could be that amygdala and hippocampus maintain the idea that the stimulus is rewarded in order to motivate the maintenance of the visual attributes of the rule and its choice by other regions. Delta/ theta activity in the amygdala and hippocampus has previously been shown to be greater when correctly rejecting a novel stimulus as previously experienced, especially when the stimuli are emotionally significant, suggesting a role in pattern separation[44]. A similar process may be involved in separating rewarded stimuli from non-rewarded stimuli in the EDID task. The process of pattern separation in the EDID task therefore appears to involve both amygdala and hippocampus simultaneously but in multiplexed frequency bands. We also found that HFG in the amygdala coded the novelty of the stimulus sets used. HFG was larger in the first block of trials after a stimulus set change, when the stimuli were novel, compared to the last block, when they were more familiar. This is consistent with the finding of responses to novel stimuli in the activities of amygdala neurons[9]. This response is believed to be reinforcing to facilitate exploration and discovery of

potential reward. It may also help establish internal representations of the stimuli which can then be used by cognitive processes or associated with reward and punishment.

The different types of responses we found in the amygdala are consistent with the multi-dimensional coding model of amygdala function derived from primate studies. In primates, amygdala neurons respond in a range of contexts including sensory, emotion, value, learning, attention, memory, decision-making and social responses[8–14,60]. This has led researchers to reject the ascription of a unitary function to the amygdala and instead apply the concept of multi-dimensional processing according to which the amygdala and its constituent neurons participate in a range of different functions which may arise from its diverse connectivity with other brain regions[15,16]. Not only does this account for the data but it would also allow the amygdala to code several different aspects of a task at the same time thereby giving rise to complex cognitive functions enabling greater flexibility and adaptability.

There are several limitations to our study. In this task we used green and red correct and incorrect words as feedback/ abstract reward and punishment. This could be potentially different to real rewards such as juice or money. Incorrect stimuli may also be punishing to subjects as it means that the task will take longer whereas correct feedback means they are making progress. However, much of human behavior is driven by abstract goals rather than low-level primary reinforcement. There could be differences between patients and normal subjects. However, patients showed similar patterns of behavior as normal subjects[5]. We were also restricted to the number of regions we can record from which are determined by clinical requirements. Given that PFC has been implicated in ED and ID shifting, it would be interesting to study interactions with amygdala and hippocampus.

The results may have applications to understanding psychiatric, neuropsychiatric and neuropsychological disorders. ED and ID can be meaningfully separated based on their underlying functional anatomical substrates as demonstrated through lesion and imaging studies translationally. ED is most commonly associated with VLPFC and to some extent DLPFC whereas ID is associated with OFC. Both ED and ID are forms of behavioral flexibility or compulsivity. In particular, ED but not ID behavior is most commonly impaired in Parkinson's disease[61] and OCD[62]. Notably, unaffected family members of patients with OCD also show impaired ED shifting highlighting its role as a cognitive endophenotype[63]. ED shifting has also been shown to be improved by deep brain stimulation targeting the subthalamic nucleus in OCD[64]. Other disorders characterized by impairments specifically in ED but not in ID include children with autism[65] and Tourette's syndrome[62]. These impairments highlight the role of cognitive inflexibility as a cognitive dimensional impairment across compulsive disorders. Our findings suggest novel sources of shifting deficits that could be tested in these populations.

The findings implicating hippocampus in rule shifting and reward learning may help understand learning and decision-making deficits after hippocampal lesions[66]. Previous fMRI studies in healthy controls and OCD did not show effects or deficits in shifting in the hippocampus[5,67]. Our findings suggest the hippocampus shifting effects may not be detected with fMRI as they were superimposed on top of a large negative, inhibitory deflection relative to baseline and the fMRI signal cannot distinguish excitatory from inhibitory activity[68]. It has previously been shown in primates that hippocampal stimulation disrupts reward learning and decision-making, perhaps because it interferes with this HFG inhibitory process[57] and activates irrelevant representations. It would be useful to investigate whether hippocampus stimulation could have beneficial effects for disorders of impulsivity and compulsivity. Our findings suggest that high gamma power may be a useful biomarker to trigger closed-loop stimulation.

In conclusion, we explored the role of the human amygdala and hippocampus in ED and ID rule shifting. Our findings suggest alternative ways of understanding these processes and their neural underpinnings consistent with recent developments in empirical and theoretical work from primates. We demonstrate that the amygdala and hippocampus are involved in using reward to learn correct rules after ED and ID shifts. Moreover, the hippocampus was involved in setting the flexibility for potential rule change and the amygdala was involved in novelty processing. These findings are consistent with current models of multidimensional processing in the amygdala and cognitive maps of task space in the hippocampus derived from primates. The findings highlight targets that could be investigated to try and understand and modify rule shifting deficits in the many disorders in which it is affected.

## Methods

### Patients

The study took place in the neurosurgical service of Ruijin Hospital, Shanghai JiaoTong University. Seventeen patients took part in total. All had severe treatment-refractory epilepsy and were undergoing stereotactic-EEG (SEEG) monitoring to locate the seizure onset zone for resection. For this reason, all patients were not taking anticonvulsive medicines at the time of testing. Eight of the patients were female and nine were male. They had a mean age of 27.6 ($SD = 8.2$) and were all right-handed. The average Montreal cognitive assessment (MOCA) score across patients was 26.1 ($SD = 2.7$). A total of 14 patients had amygdala electrodes and 13 patients had hippocampus electrodes. Each electrode had 8 2 mm contacts each separated by 1.5 mm. The ethics committee of Ruijin hospital, Shanghai JiaoTong University School of Medicine approved all procedures used. All patients provided written informed consent in accordance with the Declaration of Helsinki.

### Task

Patients completed an intra- and extra-dimensional rule switching task. This task was the same as used by Hampshire and Owen[5] except that more simplified stimuli were used to make the task easier for patients. In this task, patients are presented with two distinct shapes on either side of the screen (e.g. square and triangle). Each shape has a distinct letter overlayed on top (e.g. A and B). Shapes and letters constitute two distinct stimulus dimensions. Within each block of trials, one of the shapes or letters was the correct stimulus that had to be chosen by pressing one of two buttons with the left or right thumb corresponding to its location on the screen. This choice was repeated twice on each trial. At the second choice, the contingency between the shapes upon which the letters are superimposed is alternated. This allows us to determine which of the four stimuli the patient believes is correct. If the patient chooses the correct stimulus, they were presented with the word "Correct" in green characters. If they did not choose the correct stimulus, they were presented with the word "Incorrect" in red characters. Patients were instructed to search for the correct stimulus via trial and error and to keep choosing this stimulus until they were informed that it was incorrect. After patients had chosen the correct stimulus on three consecutive trials (six times), the stimulus that was correct was changed, requiring patients to search again for the newly correct stimulus via trial and error. There were 12 blocks in which the correct stimulus changed from letter to letter or shape to shape. This required an intra-dimensional (ID) shift or reversal. In another 12 blocks, the correct stimulus changed from shape to letter or vice-versa, thereby requiring an extra-dimensional (ED) shift of attention. After every 7 blocks of trials a new stimulus set of shapes and letters was presented. The first block, and the first blocks after stimulus set change, constituted a no shift which could be used as a baseline to compare with ED and ID shifts. There were 4 stimulus sets in total. For each stimulus set the 3 ID and 3 ED shift blocks were randomly

ordered. There were 28 blocks in total. The number of trials in each block varied depending on how quick patients were able to find the correct stimuli. The total number of trials performed ranged from 129 to 198 (M = 152, SD = 19.6). While each block necessitated either an ID, ED or no shift, the type of shift taken on each trial was computed by comparing the stimuli chosen between successive trials. On each trial, patients were presented with a fixation cross for 2–2.5 s, the choice stimuli for 2.1 s and then a cue appeared in the middle of the choice stimuli which indicated that the patient could respond. After the patient made the first response, the stimuli were presented again for 2.1 s, this time with the shape-letter combinations alternated. A response cue then appeared indicating the patient could respond. Separating the choice phases from the response phases allowed us to eliminate any potential motor confounds. The responses given at choice 1 and choice 2 were used to compute correctness and the feedback screen was presented for .6 s after patients gave their second response. Patients were instructed to respond as quickly but accurately as possible. Patients were given full instructions, demonstrations and 3 practice blocks with novel stimuli prior to completing the main task. The task was programmed and run in Matlab using Psychtoolbox functions[69].

## Electrode contact selection
The pre-implant T1-weighted MRI and post-implant CT scans were transformed into MNI ICBM152 coordinates using affine co-registration[70] in Brainstorm[71]. The MNI coordinates of the tip and trajectory of each electrode shaft was used to locate the electrode contacts. The reconstruction was overlayed on the subcortical ASEG atlas[72] to verify which contacts were located within regions of interest. Additionally, the contact positions were assessed using the Harvard-Oxford cortical and subcortical atlas. Contacts located outside of regions of interest or in regions that were subsequently resected were excluded from analysis. All contacts used for analysis are shown in Fig. 1a. We used a maximum of the first three electrode contacts from the tip for analysis as these were most frequently and precisely located within amygdala and hippocampus. In total, we analyzed 51 amygdala and 48 hippocampal contacts.

## Local field potential recording and pre-processing
Testing took place after subjects had completed their clinical assessments for seizure localization. SEEG data were recorded using a BrainAmp MR amplifier (Brain Products, Gilching, Germany) with a 1000 Hz sample rate. In addition to the SEEG electrodes, we also recorded the electro-oculogram (EOG) from electrodes placed above, below and beside the right eye. This allowed us to confirm that eye muscle activity from blinks and saccades did not contaminate the LFP data. The data were pre-processed and analyzed using Matlab 2019b and FieldTrip[73]. Offline, the data were re-referenced using a bipolar montage by subtracting adjacent contacts and notch filtered with a two-pass IIR Butterworth zero-phase lag filter to remove 50 hz powerline noise and its harmonics. The data were visually inspected (blind to conditions) to remove epochs contaminated with artefactual activity. After amplitude or power estimation, activity was averaged across channels in each region and across hemispheres to increase signal to noise ratio. The number of trials in each condition were equalized across conditions (see supplementary table 1 for trial numbers in each contrast).

## High gamma amplitude analysis
To assess high gamma modulation, we band-pass filtered the signal between 60 and 250 hz in successive 10 Hz bands using two-pass finite impulse response (FIR) zero-phase-lag filters. For each of these nineteen bands, the instantaneous amplitude/ envelope was obtained by taking the absolute value of the Hilbert transform. The amplitude was then divided by the mean activity across the entire recording and

multiplied by 100 to yield amplitude expressed as percentage of the mean. Each band was smoothed with a 250 ms moving average sliding window and then averaged together. Computing the envelope for each band separately before averaging together allowed us to avoid domination of the lower frequencies in the frequency range of the signal due to the 1/f drop off in amplitude inherent in LFP recordings[25,26]. The data were then z-scored to facilitate comparison across patients. After trials were averaged across conditions, all data were baseline corrected by subtracting the mean activity in the final 500 ms of the fixation cross period prior to first choice onset.

## Time-frequency decomposition – low frequency oscillations
Time-frequency decomposition was performed using multi-taper convolution. For each trial, the data were windowed using a sliding time-window centered at 20 ms increments and tapered to reduce spectral leakage before calculating power. We analyzed logarithmically spaced frequencies between 2 and 32 Hz at 25 scales per octave using a single hanning tapered time-window with a duration of 6 cycles. The time-frequency representations were averaged across conditions and baseline corrected by calculating percent signal change from −500 ms to 0 ms prior to the onset of the cue during the fixation period ((Active-Baseline)/Baseline*100). We used percent signal change as it corrects for the 1/f drop off in signal amplitude as frequency increases, thereby allowing us to compare different low frequency bands.

## Phase-locking values
Phase-locking analysis (PLV/PLS) was used to test for oscillatory synchronization between regions. PLV is a measure of the consistency of the phase θ differences between oscillations at two channels (x and y) at a particular time t and frequency f across trials, irrespective of absolute phase and amplitude[39]:

$$PLV(t,f) = \frac{1}{Ntrials}\left|\sum_{n=1}^{n=Ntrials} \exp(i[\theta n, x(t,f) - \theta n, y(t,f)])\right|$$

The resulting coefficient is bound between 0 and 1 indicating the strength of PLV. PLV was calculated using the same time-frequency decomposition parameters as used for the low-frequency analysis. This analysis was performed on 11 patients that had both amygdala and hippocampal electrodes. In total, PLV was calculated between 78 pairs of amygdala and hippocampus electrodes. All electrodes were paired within each hemisphere.

## LFP statistical analysis
Statistical significance of differences in LFP activity between conditions were evaluated using non-parametric cluster-based permutation testing as it allows for the control of multiple comparisons and does not assume that the data are normally distributed[74]. The permutation test works by repeatedly permuting the mapping between condition labels and time-series, calculating paired samples t-statistics for each data point, clustering datapoints that exceed a *p*-value of .05, and extracting the sum of t-values that form the largest cluster to build a distribution which can be used to compare with the non-permuted clusters. If the non-permuted cluster statistic is larger than 95% of clusters obtained after permuting the data (i.e. *p* < .05), it is considered significant. We used all possible permutations, which was 2048, 4096, 8192 and 16384 for contrasts with 11, 12, 13 and 14 patients, respectively. We only report clusters as significant if the cluster-level significance exceeded a Bonferroni correction for the number of regions tested which was two (*p* < .025). To test for general increases or decreases in LFP activity against baseline, we performed one sample Wilcoxon signed ranks tests on the baseline corrected data and corrected *p*-values for multiple comparisons over time using false discovery rate (FDR) correction[75]. Time points were only considered significant if corrected *p*-values were below the Bonferroni correction

for the number of regions tested ($p < .025$). To test for differences between regions, we subtracted conditions and compared the differences between amygdala and hippocampus using an independent samples t-test with cluster correction using 5000 permutations or a Wlicoxon rank sum test on averaged time windows. Effect sizes were estimated by calculating Cohens D on activity averaged within significant clusters. See Sawilowsky[76] for definitions of effect size strength.

## Binary support vector machine classification

We tested if we could use patterns of activity across all time points to classify condition membership of individual trials within and across regions. To do this, we trained binary support vector machines (SVM) on our data (trials * time) using the fitcsvm function in Matlab. Within regions, accuracy was assessed using a leave-one-out design whereas between regions it was assessed by training on amygdala and testing on hippocampus data and vice versa. Training and testing between regions was performed on 11 patients who had both amygdala and hippocampal electrodes. Percentage correct classification was calculated for each patient and tested for significance using a one-sample, Wilcoxon signed ranks test against chance (50%). One-tailed p-values were used as significant accuracy would be above chance only. Bonferroni-Holm correction based on four contrasts for each training and test region combination was used to correct for multiple comparisons.

## Behavioral data analyses

Reaction times were normalized using logarithmic transformation and z-scored to facilitate comparison across patients. Reaction times were analyzed with a linear mixed effects (LME) model using the fitlme function in Matlab with the restricted maximum likelihood method. This method is advantageous over standard t-tests as it allows us to model trial-by-trial variation as a fixed effects factor, inter-subject variation as a random-effects factor and is not biased by differences in the number of trials between conditions. We confirmed that the assumptions for LME analysis were met. The residuals were shown to be normally distributed using histograms and qq-plots and heteroskedastic by plotting them as a function of the fitted values. As errors are binomial, they were analyzed using a generalized linear mixed effects model (GLME) with a logit link function using the fitglme function in Matlab. The significance of each factor in the LME and GLME models are evaluated using a t-statistic.

## Reporting summary

Further information on research design is available in the Nature Portfolio Reporting Summary linked to this article.

## Data availability

All data is provided within the article, supplementary materials and Source data file. Further information regarding the findings are available upon request to the corresponding author(s). Source data are provided with this paper.

## Code availability

Code used in the study is available at: https://github.com/VoonLab/Reward_recalibrates_NatComms_Code.git.

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

## Acknowledgements

We would like to thank all the patients for taking part. This work was supported by the following grants: Natural Science Foundation of China grant (81771482) to BMS; SJTU Trans-med Awards Research (2019015) to BMS; Shanghai Clinical Research Centre for Mental Health (19MC191100) to BMS; Medical Research Council Senior Clinical Fellowship (MR/W020408/1) to VV; National Natural Science Foundation of China Grant (T2250710686) to VV; STI 2030 – Major Projects Grant (2021ZD0200407) to VV. This paper is in memory of Ms Ruoqi Yang. Without her hard work and dedication to patient testing this study would not have been possible.

## Author contributions

L.M. and V.V. designed research; L.M. analysed data; L.M. wrote paper; L.M., Q.D., Y.F., R.Y., W.L., B.S. and S.Z. performed research.

## Competing interests

The authors declare no competing interests.
