## [Transparent Peer Review file · Nature Communications]

Reward recalibrates rule representations in human amygdala and hippocampus intracranial recordings

Corresponding Author: Dr Luis Manssuer

Version 0:

Reviewer comments:

Reviewer #1

(Remarks to the Author)

Summary:

Here the authors leverage a relatively rare clinical opportunity to record local field potential activity in humans undergoing intracranial monitoring for epileptic seizure localization. The Wisconsin card sorting task has been widely used in both humans during fMRI or monkeys during intracranial recordings. Here, the authors conducted intracranial recordings in many participants (N=17) during a variation on the Wisconsin card sorting task, the extra-dimensional and intra-dimensional task (the EDID task). Classically, recordings during this task are done in the lateral prefrontal cortex. One of the most interesting aspects of the paper is that recordings were conducted in regions of the brain (the hippocampus and amygdala) that are not known to be relevant for this task. Indeed, the authors found that activity in these regions is correlated with win-stay lose-shift decisions.

Major comments:

1. While the authors demonstrated that hippocampal and amygdala activity correlates with various features of task-related behavior, I do not think the findings are very surprising. Additionally, the authors did not explicitly state whether they were testing the hypothesis that hippocampal and amygdala activity play a role in decision-making during this task.
2. For claims regarding the difference in activity between the hippocampus and amygdala, can the authors conduct a statistical test to compare activity across the two regions? A significant finding in the amygdala and an insignificant finding in the hippocampus does not prove that activity in these two regions are different (please use a statistical test to directly compare difference between hippocampus and amygdala activity).

Minor comments:

1. A lot of the ideas in the discussion are really interesting and maybe should be included in the introduction. I think moving some of the content could help frame the paper a little better. For example, the context that most previous work has focused on the PFC.
2. For all the plots showing effects across patients ordered by size of effect (like in 2D), it might be more useful to show the data in boxplots with a scatter of the individual data points (like in 1C-D). All of the plots (like 2D) have different y axis scales, and it's hard to see the mean. The colors on these plots don't have a true meaning as the same color represents multiple patients, and the patients are reordered in every plot.
3. Can the authors please describe how the EDID task is different from the classic Wisconsin card sorting task.
4. Since Figure 6 is meant to be a summary figure, can the authors provide more clarifying information on what is going on in the figure itself specifically for Figure 6A? (e.g., on the figure say which scenario is the top vs. bottom, and indicate which the data supports). Instead of including a visual of the theta oscillation, you may want to include a shorthand for rule updating (as a parallel to win-stay lose-shift). It was unclear to me at first that both theta and high gamma were important for the reward component of Figure 6A (bottom).
5. Could the authors consider using different colors for legend items that mean different things? Red and blue are primarily used across the manuscript and mean different things throughout.
6. If feedback is presented for 0.6 seconds, how do the authors interpret feedback signals that only achieve significance around the ~2 second mark?

Reviewer #2

(Remarks to the Author)

In this paper the authors explore the role of the amygdala and hippocampus in interdimensional and extradimensional rule-shifting to determine the role of these regions in reversal learning and attention shifting. They use iEEG recordings from 17 patients with epilepsy who have electrodes implanted in the hippocampus and amygdala for the purpose of discovering seizure onset zones. The authors use gamma power as a proxy of population level spiking. They took a regional approach, looking at gamma power -- high frequency activity (HFA)-- in both the amygdala and hippocampal regions separately at different timepoints in the switching task (reward phase, decision phase). They found that the amygdala showed win stay -- lose shift signals in response to feedback. There was increased HFA to correct feedback on shift trials relative to no shift trials, and increased HFA activity in response to incorrect relative to correct feedback on no-shift trials. During the decision phase, they found a significant increase in theta activity on no-shift trials relative to shift trials throughout the trial. They also found a novelty response -- increase HFA in response to the choice stimuli on the first trial after stimulus set change. The authors assert that the amygdala is engaging in reinforcement learning by updating goals dependent on both reward and punishment information (reinforcement value representation). They state that the theta activity is a working memory representation important for decision-making processes across trials. They also suggest that the amygdala response to novelty may reflect the formation of stimulus memories, particularly in the basolateral region. In contrast, the hippocampus showed a win-stay signal, but was otherwise inhibited during the task. There was an increase in HFA on correct shift relative to correct no-shift trials and correct vs incorrect shift trials. They also found HFA on correct vs incorrect trials in the intertrial period. During the decision phase, they found decreased HFA activity across all trials, with a greater decrease on no-shift trials relative to shift trials. The authors assert that the hippocampus is updating the correct rule in memory based on reward alone, maintaining the correct rule in memory, and preventing interference between competing representations of the rule.

Overall, the paper is well written, and the findings are novel. The methodology for the analyses presented is sound. For publication in this journal, I would recommend that the authors include additional analyses to help prove the assertions they are making about the role of the amygdala and hippocampus in reinforcement learning. They could show that brief electrical stimulation of contacts within the amygdala or hippocampus during the reward/decision period disrupts value representation encoding or rule maintenance in the predicated manner. They could build a predictive model of behavior from one brain region and test it on the second brain region (win-stay behavior for example), or a second task.

They should also consider analyses beyond spectral power. They could extend their regional analyses to also include a network analysis. They could consider looking at coherence or phase-based connectivity metrics between the HPC and amygdala, or between the amygdala/HCP and other regions such as the PFC if electrodes present in a subset of participants. If available, they could analyze DTI imaging to identify regional specificity of electrode contacts perhaps in relation to white matter tracts. For example, they mention the potential importance of the basolateral amygdala in novelty detection, which could be directly tested.

Reviewer #3

(Remarks to the Author)

This paper is compelling and provides important new data regarding activity within human mesial temporal lobe that likely contributes to behavioural flexibility as evidenced experimentally in performance of intra- and extra- dimensional shifts. The paper uses intracranial recording of local field potentials (LFPs) in patients with epilepsy, extracting changes in high frequency gamma HFG and theta bands to appraise neural correlates of experimental challenges and behavioural response. It is inferred that both amygdala and hippocampus neural activity encode reward signals that are modulated by the learning rules of the task with evidence for attentional dependency. Differences in the regional responses to positive and negative feedback were shown, providing more detailed insight into the distinct contributions of amygdala and hippocampus the changing contingent value of environmental stimuli. This work is important and novel. It provides sting insight into neural substrates of cognitive processes with clinical relevance's, extending what can be inferred from neuroimaging to more proximate measures of neuronal circuit function in humans.

My comments are minor and relate mostly to improving the abstract, providing of more motivation and detail about the High Frequency Gamma and related measures, and enhancing aspects of presentation.

Abstract

The abstract could be improved for greater clarity:

It implies that there have been few human studies other than fMRI in human reversal / set-shifting (despite many lesion studies and some stimulation /inhibition studies.

The comments in relation to monkey single unit studies underplays inferences from past studies dating to 1970s and some evidence again from lesions.

The mode of direct recording of human neural activity (subfrequency bands of LFPs) isn't mentioned in the abstract, perhaps implying that (arguably more direct) single unit spiking measures were conducted in humans also. The results are somewhat glossed over in favour of early presentation of interpretation of findings. The last line concerning biomarkers of set shifting deficit in neuropsychiatric disorder seems convoluted. Certainly the findings help the understanding of mechanisms but the rationale for what these regional correlates might add in terms of biomarker function is not addressed in the paper.

Introduction

This is generally clear and well written.

Review of reversal learning and single unit recording in primates; should include mention of Thorpe et al 1983 (OFC), and Sanghera et al., 1979 (amygdala), studies which contributed to longstanding proposal by Rolls and others (e.g. Wilson and Rolls 2005) that rapid reversal learning for adaptive behaviour is supported in OFC, while relative inflexibility of learned associations encoded in amygdala activity (and advantageous over longer time periods, emerging e.g. in extinction reinstatement). This is reflected also in Morris and Dolan fear reversal imaging studies in humans

The selection of high frequency gamma as an a priori measure regional (population-level)s spiking is needs expanding in terms of motivation (i.e. spelling out to explain the 2 references); Later in the paper, the concept of 'inhibitory activity' and results pertaining to another LFP band (theta) are introduced. These need earlier explanation in the introduction. More background on HFG is warranted in relation to known representation of information and whether all frequency bands were explored.

Results

The behavioral results are clear

In LFPs section the terms 'HFG signals' and 'HFG activity' is used when perhaps power is meant. Statement about attention when interpreting finding need unpacking more (e.g. page 9).

The main issue is the reporting of these HFG /LFP results is the use P values as proxy of effect sizes. I suggest this is changed.

The switch to analyses of theta frequency changes on page 12 needs better rationale.

Other areas of the paper are satisfactory / interesting and make valuable and novel points. Greater clarity could be made in the discussion to what ED/ID has relevance to psychopathology (i.e. what are deeper links to symptoms e.g. perseverative cognition, repetitive behavior in context of conditions including OD but possibly other others (autism, eating disorder tic disorder, psychoses, neurodegeneration) and actually how ED/ID distinction is relevant /meaningful

Overall, the paper is strong, though there are opportunities for improvement in presentation

Version 1:

Reviewer comments:

Reviewer #1

(Remarks to the Author)

Thank you for the revised manuscript and detailed responses. All of my concerns have been addressed with the additional analyses and manuscript edits.

Reviewer #2

(Remarks to the Author)

The authors have addressed my comments in a satisfactory manner. They added predictive modeling of behavior from the AMY and HPC LFP signals. They also added phase-amplitude coupling to provide additional information on the communication between the AMY and HPC. Their findings, which suggest that communication between the two regions is state dependent, is interesting and novel. The improvements in the figures, interpretative text, and addition of effect size to the results also strengthened this paper. I believe that this is an interesting paper suitable for publication in the journal.

Reviewer #3

(Remarks to the Author)

The authors have addressed the points I raised to my satisfaction.

Revisions to the manuscript in response to all the reviewers comments appear to have enhanced the readability and accessibility of this interesting dataset.

The methods and conclusions are appropriate and clear.

I do not have further comments

REVIEWER COMMENTS

Thank you for your very helpful and constructive insights. The paper is much stronger for it. We believe we have addressed all comments fully and hope that our amendments are satisfactory. However, we are more than happy to make further adjustments to fine tune the paper if required. All responses to the reviewers are written in red font below each point and changes to the manuscript are also in red font.

Reviewer #1 (Remarks to the Author):

Summary:

Here the authors leverage a relatively rare clinical opportunity to record local field potential activity in humans undergoing intracranial monitoring for epileptic seizure localization. The Wisconsin card sorting task has been widely used in both humans during fMRI or monkeys during intracranial recordings. Here, the authors conducted intracranial recordings in many participants (N=17) during a variation on the Wisconsin card sorting task, the extra-dimensional and intra-dimensional task (the EDID task). Classically, recordings during this task are done in the lateral prefrontal cortex. One of the most interesting aspects of the paper is that recordings were conducted in regions of the brain (the hippocampus and amygdala) that are not known to be relevant for this task. Indeed, the authors found that activity in these regions is correlated with win-stay lose-shift decisions.

Major comments:

1. While the authors demonstrated that hippocampal and amygdala activity correlates with various features of task-related behavior, I do not think the findings are very surprising. Additionally, the authors did not explicitly state whether they were testing the hypothesis that hippocampal and amygdala activity play a role in decision-making during this task.

We state that we are investigating ED and ID rule shifting which involves multiple processes including decision-making. We have made this more explicit in the introduction.

See page 5.

2. For claims regarding the difference in activity between the hippocampus and amygdala, can the authors conduct a statistical test to compare activity across the two regions? A significant finding in the amygdala and an insignificant finding in the hippocampus does not prove that activity in these two regions are different (please use a statistical test to directly compare difference between hippocampus and amygdala activity).

Our main claim regarding distinctions between regions centred around the difference between correct and incorrect in the no shift condition in the amygdala. This is interpreted as a lose-shift signal which is absent in the hippocampus and therefore we infer that the amygdala is more involved in reinforcement learning. In order to substantiate this claim, we extracted average activity within the significant time interval in both regions, subtracted the difference between conditions and compared

the two regions with a Wilcoxon rank sum test (between-subjects statistic was used due to some patients having amygdala but not hippocampus electrodes and vice versa and non-parametric test was used in case of non-normality of HFG). Indeed, the difference between incorrect and correct in the no shift condition was significantly larger in the amygdala compared to the hippocampus ($p=.022$). We also found that we could use outcome activity to predict correct and incorrect no shift trials using an SVM classifier. However, predictive ability was not found in the hippocampus and did not generalize across regions. We have included this in the results. We also directly compared amygdala and hippocampus activity in the decision phase and found that the hippocampus was more inhibited compared to the amygdala as would be expected. We have added this to the results and the figure to the supplementary materials. Note that we used boxed plots to show average activity across patients as it was not a within-subjects test.

In the revised paper, we have also included extra analyses which relate to this issue such as the training and testing within and between regions using SVMs and connectivity analyses.

Most of the effects were present to some degree in both amygdala and hippocampus. However, effects tended to vary in their strength between regions, which makes it difficult to make dichotomies. This would be expected given the regions are fairly close and interconnected. It might be best to think of activity as having a functional gradation across the two regions. While not completely absent in amygdala vs hippocampus, one region tends to take precedence.

We originally aimed only to report the strongest, most robust and reliable effects, but we have also mentioned some of the trending effects which pertain to this issue such as low frequency activity in the hippocampus and HFG novelty effects in the hippocampus.

See page 12-16, 18-19. See Figure 4 and 8. See supplementary figures 4 and 5.

Minor comments:

1. A lot of the ideas in the discussion are really interesting and maybe should be included in the introduction. I think moving some of the content could help frame the paper a little better. For example, the context that most previous work has focused on the PFC.

We have mentioned that most work has focused on the PFC in the intro. The intro has been expanded. Hopefully, the extra information we have included will help frame the paper better.

See pages 3-6.

2. For all the plots showing effects across patients ordered by size of effect (like in 2D), it might be more useful to show the data in boxplots with a scatter of the individual data points (like in 1C-D). All of the plots (like 2D) have different y axis scales, and it's hard to see the mean. The colors on these plots don't have a true meaning as the same color represents multiple patients, and the patients are reordered in every plot.

We have kept the original format of the plots but have included a horizontal line that indicates the mean difference and have changed the colors to be uniform across patients. We have done this as readers will not be interested in any single patient and so the colors are irrelevant. We found these graphs show the effects much clearer. Using condition differences removes any large differences between patients that are common to both conditions and might make it more difficult to see condition differences. This is one advantage of using a within-subjects design. We have also changed the y axis limits so that they are all the same scale for comparable contrasts.

3. Can the authors please describe how the EDID task is different from the classic Wisconsin card sorting task.

The ED trials of the EDID task should theoretically capture similar processes as rule shifts in the WCST (See Mansouri et al., 2020). However, there are several notable differences between the two tasks. The main difference is that the WCST confounds ED and ID shifting when there is a rule change indicated by changes in reinforcement. There are several other differences. Firstly, the WCST may have more than two dimensions. For example, there may be shape, color and number in the WCST where as in the EDID task there are only two dimensions (Letter and Shape). Therefore, there may be extra processes involved in selecting a dimension in the WCST which are not in the EDID. The other main difference is that ED shifts in the EDID task are defined based on the patients own behavior relative to the previous trial instead of just being determined by the experimenter. Another difference is that in addition to not shifting, in the EDID task patients also have the alternative to complete an ID shift, so there may be some processes involved in the EDID task that involve arbitrating between the two types of possible shift. In the WCST, the stimuli are multidimensional, whereas in the EDID task, they are compound (overlayed on top of each other). The spatial position of the stimuli are also different. Typically, in the WCST, the matching stimulus is in the top centre of the screen and the choice stimuli are underneath. In contrast, the stimuli in our task are presented on the left and right side of the screen.

We have added to the introduction more clarity of the benefits of the EDID task relative to the WSCT.

See pages 5 and 7.

4. Since Figure 6 is meant to be a summary figure, can the authors provide more clarifying information on what is going on in the figure itself specifically for Figure 6A? (e.g., on the figure say which scenario is the top vs. bottom, and indicate which the data supports). Instead of including a visual of the theta oscillation, you may want to include a shorthand for rule updating (as a parallel to win-stay lose-shift). It was unclear to me at first that both theta and high gamma were important for the reward component of Figure 6A (bottom).

Based on your feedback, we have decided to omit the top part of the figure to keep it much simpler and easy to understand. We removed the theta graphic and replaced it

with “rule updating”. We have included text to make it more explicit what we are trying to convey. Our original graph was meant to show that instead of win-stay-lose-shift signals occurring in an obligatory, bottom-up manner, they only occur at transitions between bouts of shifting and staying trials which may be controlled by attention in a top-down fashion. Hopefully, this is conveyed much clearer with the changes we have made.

See figure 9.

5. Could the authors consider using different colors for legend items that mean different things? Red and blue are primarily used across the manuscript and mean different things throughout.

We have done this. See figures 2-8.

6. If feedback is presented for 0.6 seconds, how do the authors interpret feedback signals that only achieve significance around the ~2 second mark?

The presentation of feedback is relatively brief (.6 seconds). To perform the task successfully it is critical that patients remember the rule in between trials (patients need to get three trials correct in sequence to progress to the next block). We believe that the increased hippocampal activity after reward, relative to punishment, in the ITI, reflects maintenance, rehearsal or consolidation of the rule in memory, so patients don't forget the rule in between trials. These types of rule working memory signals have also been demonstrated in other brain areas (e.g. DLPFC) in macaques during the WCST (See Mansouri et al. 2020). It is interesting that this effect ramps up gradually from the point of correct feedback and becomes greater over time (see supplementary figure 3) perhaps because a stronger memory trace was needed as time passed to prevent decay or due to increased preparation for the upcoming trial. Recent studies have also demonstrated ramping dynamics of high gamma activity in intracranial recordings during decision-making (Weber et al., 2024; Xie et al., 2024). There may be several components in the HFG signal that occur early and late in response to the stimulus. The stimulus presented in the ITI is a neutral fixation cross which is exactly the same for all conditions. Therefore any differences between conditions must be due to carry over from the feedback presentation or active processes in preparation of the upcoming trial based on previous information obtained from the feedback. It is also noteworthy, that a previous study has demonstrated that high gamma activity in the OFC can be sustained to contain reward-related information acquired on previous trials to influence behavior on subsequent trials (Saez et al., 2018). Furthermore, OFC is also believed to be involved in cognitive mapping of task space via interactions with the hippocampus (e.g. Knudsen & Wallis, 2020). We have added this to the discussion.

See pages 12-13 and 24-25. See supplementary figure 3.

Reviewer #2 (Remarks to the Author):

In this paper the authors explore the role of the amygdala and hippocampus in interdimensional and extradimensional rule-shifting to determine the role of these regions in reversal learning and attention shifting. They use iEEG recordings from 17 patients with epilepsy who have electrodes implanted in the hippocampus and amygdala for the purpose of discovering seizure onset zones. The authors use gamma power as a proxy of population level spiking. They took a regional approach, looking at gamma power -- high frequency activity (HFA)-- in both the amygdala and hippocampal regions separately at different timepoints in the switching task (reward phase, decision phase). They found that the amygdala showed win stay – lose shift signals in response to feedback. There was increased HFA to correct feedback on shift trials relative to no shift trials, and increased HFA activity in response to incorrect relative to correct feedback on no-shift trials. During the decision phase, they found a significant increase in theta activity on no-shift trials relative to shift trials throughout the trial. They also found a novelty response – increase HFA in response to the choice stimuli on the first trial after stimulus set change. The authors assert that the amygdala is engaging in reinforcement learning by updating goals dependent on both reward and punishment information (reinforcement value representation). They state that the theta activity is a working memory representation important for decision-making processes across trials. They also suggest that the amygdala response to novelty may reflect the formation of stimulus memories, particularly in the basolateral region. In contrast, the hippocampus showed a win-stay signal, but was otherwise inhibited during the task. There was an increase in HFA on correct shift relative to correct no-shift trials and correct vs incorrect shift trials. They also found HFA on correct vs incorrect trials in the intertrial period. During the decision phase, they found decreased HFA activity across all trials, with a greater decrease on no-shift trials relative to shift trials. The authors assert that the hippocampus is updating the correct rule in memory based on reward alone, maintaining the correct rule in memory, and preventing interference between competing representations of the rule.

Overall, the paper is well written, and the findings are novel. The methodology for the analyses presented is sound. For publication in this journal, I would recommend that the authors include additional analyses to help prove the assertions they are making about the role of the amygdala and hippocampus in reinforcement learning.

They could show that brief electrical stimulation of contacts within the amygdala or hippocampus during the reward/decision period disrupts value representation encoding or rule maintenance in the predicated manner. They could build a predictive model of behavior from one brain region and test it on the second brain region (win-stay behavior for example), or a second task.

Our task is not suitable for stimulation. This is because, if stimulation disrupted performance, the task would continue indefinitely as correct performance is necessary for the task to advance. Furthermore, we would need many more new participants which is difficult due to the rarity of the patients and amount of time needed to recruit and test. Standard WCST or reversal tasks would be better suited to address this question but then you not be able to directly compare ED and ID as well as you could with our EDID task. Stimulation studies of the hippocampus have

been performed in reversal tasks (Knudsen & Wallis, 2020) in primates. We refer to this in the paper.

See page 27.

However, we can build a predictive model of behavior from one region and test on a second region. We did this by training and testing binary support vector machines (SVMS) on the LFP data. This analysis provided additional interesting and informative results. We found that we could use SVMs on the HFG signal to significantly predict condition membership across several contrasts both within and between amygdala and hippocampus. We have added these results to the paper.

See pages 13-15 and 23.

They should also consider analyses beyond spectral power. They could extend their regional analyses to also include a network analysis. They could consider looking at coherence or phase-based connectivity metrics between the HPC and amygdala, or between the amygdala/HCP and other regions such as the PFC if electrodes present in a subset of participants. If available, they could analyze DTI imaging to identify regional specificity of electrode contacts perhaps in relation to white matter tracts. For example, they mention the potential importance of the basolateral amygdala in novelty detection, which could be directly tested.

We have now considered analyses beyond spectral power. We have extended our regional analyses to also include a network analysis. We have looked at phase based connectivity between the amygdala and hippocampus. We found phase-locking between amygdala and hippocampus in response to feedback which dovetails nicely with the findings in the high gamma range. We show that phase-locking values (PLV) between amygdala and hippocampus in the delta-theta range are greater on correct shift trials relative to correct no shift trials. We previously showed that high gamma differentiates these conditions in both the amygdala and hippocampus. The PLV findings are a significant addition to our understanding of the underlying neural mechanisms in that they suggest that the two regions may be synchronized or linked via PLV in the delta-theta band. It also adds to our understanding of the conditions under which delta-theta PLV to reward occurs. We previously showed delta PLV between amygdala and hippocampus during reward receipt (Manssuer et al., 2022). Our new findings suggest that this is not obligatory and dependent on the goals and attentional set of the patient. We only focused on the connectivity between amygdala and hippocampus as these are very commonly targeted regions and therefore there are a sufficient number of electrode pairs for a reliable analysis.

We have fully explored our data, looking at a range of different features including power (or amplitude), phase, phase-amplitude coupling, coherence and phase-locking values. We did not find as strong effects as we did with the high gamma analysis. Therefore we decided to focus on the most robust results in the high gamma range. In addition to high gamma activity being the most robust, we also feel it is the most interesting theoretically, given both past research based on its role in reward (Lopez-Persem et al., Guegen et al., Saez et al.) and information processing and studies which have shown its physiological relationships with single unit activity

(Buzsaki et al., 2012). It therefore provides useful parallels with single unit recordings from animals. In contrast, while low frequency phase may play a role in regulating high frequency activity and synchronising regions (Canolty & Knight, 2006), it is not as directly involved in local information processing as high frequency activity. We have quite a lot of data in the high gamma range and therefore this takes up most of the room. Many studies of a similar nature focus solely on high frequency activity for these reasons (e.g. Saez et al., 2018, Manssuer et al., 2023).

We have looked at correlations of LFP activity with X, Y and Z MNI coordinates to examine any regional distinctions but we did not find any associations. The electrodes are implanted by clinicians for clinical purposes on a case by case basis and so the coverage of the amygdala is not systematic within or across individuals. Therefore there is no guarantee that all parts of the amygdala or hippocampus will be sampled to a sufficient extent. This makes it difficult for analyses of regional specificity based on SEEG recordings. Even with many patients, there may be intersubject variation confounded with electrode positions. This type of analysis is better suited for ECoG recordings or animal studies where there is more control over positioning of electrodes.

See pages 15-16 and 23. See figure 5.

Reviewer #3 (Remarks to the Author):

This paper is compelling and provides important new data regarding activity within human mesial temporal lobe that likely contributes to behavioural flexibility as evidenced experimentally in performance of intra- and extra- dimensional shifts. The paper uses intracranial recording of local field potentials (LFPs) in patients with epilepsy, extracting changes in high frequency gamma HFG and theta bands to appraise neural correlates of experimental challenges and behavioural response. It is inferred that both amygdala and hippocampus neural activity encode reward signals that are modulated by the learning rules of the task with evidence for attentional dependency. Differences in the regional responses to positive and negative feedback were shown, providing more detailed insight into the distinct contributions of amygdala and hippocampus the changing contingent value of environmental stimuli. This work is important and novel. It provides sting insight into neural substrates of cognitive processes with clinical relevance's, extending what can be inferred from neuroimaging to more proximate measures of neuronal circuit function in humans.

My comments are minor and relate mostly to improving the abstract, providing of more motivation and detail about the High Frequency Gamma and related measures, and enhancing aspects of presentation.

Abstract

The abstract could be improved for greater clarity:
It implies that there have been few human studies other than fMRI in human reversal / set-shifting (despite many lesion studies and some stimulation /inhibition studies.

We have tried to clarify this. See page 2.

The comments in relation to monkey single unit studies underplays inferences from past studies dating to 1970s and some evidence again from lesions.

This is a very good point. We have omitted the term “recent” from the description of the single unit studies in the abstract and throughout the rest of the paper. We will also reference these papers more in the manuscript. See page 2, 4-5.

The mode of direct recording of human neural activity (subfrequency bands of LFPs) isn't mentioned in the abstract, perhaps implying that (arguably more direct) single unit spiking measures were conducted in humans also. The results are somewhat glossed over in favour of early presentation of interpretation of findings. The last line concerning biomarkers of set shifting deficit in neuropsychiatric disorder seems convoluted. Certainly the findings help the understanding of mechanisms but the rationale for what these regional correlates might add in terms of biomarker function is not addressed in the paper.

We have explicitly stated we have used the method of local field potentials in the abstract and have mentioned which frequency bands differences between conditions were found in. We have also separated the results from our interpretation of the results. We have re-worded our conclusions to exclude the statement about biomarkers.

See page 2, 4-6.

Introduction

This is generally clear and well written.

Review of reversal learning and single unit recording in primates; should include mention of Thorpe et al 1983 (OFC), and Sanghera et al., 1979 (amygdala), studies which contributed to longstanding proposal by Rolls and others (e.g. Wilson and Rolls 2005) that rapid reversal learning for adaptive behaviour is supported in OFC, while relative inflexibility of learned associations encoded in amygdala activity (and advantageous over longer time periods, emerging e.g. in extinction reinstatement). This is reflected also in Morris and Dolan fear reversal imaging studies in humans.

There have been single unit studies of reversal learning in the macaque amygdala, but we are interested in both ID and ED shifting and the comparison between the two. These types of studies focusing on the amygdala and hippocampus are much less frequent.

Our main results for amygdala are in the outcome phase whereas I believe the reversal learning literature mostly focuses on responses to the discriminative stimuli (i.e. the decision phase) rather than outcomes. Our task also uses a two-choice task (or rather four choice task if including ED) which may be different to the above studies as patients may retrieve the value of reward associations with S+ and S- within the same trial in order to make a decision. It is also worth noting that our task is somewhat different to the reversal learning studies used in macaques in that in our task the ID and ED conditions are defined based on the patients own choices relative to the previous trial, rather than being set by the experimenter. Nevertheless, we have expanded the introduction to include mention of these references and caveats.

See pages 4-5, 7.

The selection of high frequency gamma as an a priori measure regional (population-level)s spiking is needs expanding in terms of motivation (i.e. spelling out to explain the 2 references); Later in the paper, the concept of 'inhibitory activity' and results pertaining to another LFP band (theta) are introduced. These need earlier explanation in the introduction. More background on HFG is warranted in relation to known representation of information and whether all frequency bands were explored.

We have expanded the introduction to give more details about HFG and the information it has been shown to encode. We have mentioned that HFG can provide information about task-induced activation and suppression (i.e. inhibition) of neural activity which is difficult to obtain from other measures (as detailed by Lachaux et al., 2012). We have stated all the frequency bands the results were found in.

See pages 2, 4-6.

Results

The behavioral results are clear

In LFPs section the terms 'HFG signals' and 'HFG activity' is used when perhaps power is meant. Statement about attention when interpreting finding need unpacking more (e.g. page 9).

This is a good point although it's the reverse. We have changed any mention of HFG power to HFG activity or signals. The reason for this is that in our particular method of spectral decomposition, we have bandpass filtered the data and applied the Hilbert transform. Therefore activity is expressed as amplitude rather than power. Although power can be easily obtained by squaring the signal, we prefer amplitude because power is more skewed and hence deviates from normality (although most of our analyses do not depend on normality). Power is typically obtained after an FFT or Morlet wavelet transform where squaring is necessary to the calculation but many people square root the data so that it is not skewed for subsequent analysis and display (Burgess, 2019; Kiebel, Tallon-Baudry & Friston, 2005). Either way, amplitude or power are equally valid and should not make much difference to the results. We have made sure that this terminology is consistent throughout the paper and have changed all references to HFG power. Information about the transform is provided in the methods. For low frequency analysis we used the multi-taper method which outputs power. Therefore power is used to describe low frequency activity.

We have added much more detail at this part of the results to make our ideas about attention clearer to readers.

See pages 10-11.

The main issue is the reporting of these HFG /LFP results is the use P values as proxy of effect sizes. I suggest this is changed.

For the LFP analyses, we have included Cohens D estimates of effect sizes in addition to p-values. For the behavioral data analyses, we have added the coefficient

value and its 95% confidence intervals.

The switch to analyses of theta frequency changes on page 12 needs better rationale.

We did not have any a priori prediction about which frequency bands would show differences between conditions. This is why we performed a cluster analysis over all of time and frequency space so we could localize activity differences in a data driven manner. Therefore, we do not begin this section by talking about theta specifically. Even if we did know that previous studies found an effect in the theta band, there can still be variability between individuals and studies in the exact timing and frequency of differences between conditions which is why time frequency analysis is useful. It also allows us to investigate several frequencies simultaneously. Instead, we have added more introduction about how low frequency oscillations are involved in cognitive processes. After we have mentioned the results in the theta band, we then proceed to discuss what the theta results may represent. We have added some extra discussion of what the theta signal may represent in relation to previous studies.

In addition, in the introduction, we have mentioned more about the distinction between high and low frequency activity and what they reflect. We introduce low frequency time-frequency analysis and the distinction between different low frequency bands in the section on PLV which occurs before the section on low frequency power.

See pages 4-5, 15-16, 18-19, 25.

Greater clarity could be made in the discussion to what ED/ID has relevance to psychopathology (i.e. what are deeper links to symptoms e.g. perseverative cognition, repetitive behavior in context of conditions including OD but possibly other others (autism, eating disorder tic disorder, psychoses, neurodegeneration) and actually how ED/ID distinction is relevant /meaningful

Overall, the paper is strong, though there are opportunities for improvement in presentation

We have added this to the discussion:

“The results may have applications to understanding psychiatric, neuropsychiatric and neuropsychological disorders. ED and ID can be meaningfully separated based on their underlying functional anatomical substrates as demonstrated through lesion and imaging studies translationally. ED is most commonly associated with VLPFC and to some extent DLPFC and ID with OFC. Both ED and ID are forms of behavioural flexibility or compulsivity. In particular, ED but not ID behaviour is most commonly impaired in Parkinson’s disease⁶¹ and OCD⁶². Notably, unaffected family members of patients with OCD also show impaired ED shifting highlighting its role as a cognitive endophenotype⁶³. ED shifting has also been shown to be improved by deep brain stimulation targeting the subthalamic nucleus in OCD⁶⁴. Other disorders characterized by impairments specifically in ED but not in ID include children with autism⁶⁵ and Tourette’s syndrome⁶². These impairments highlight the role of

cognitive inflexibility as a cognitive dimensional impairment across compulsive disorders. Our findings suggest novel sources of shifting deficits that could be tested in these populations.”

See pages 26-27.

References

Burgess A. P. (2019). How Conventional Visual Representations of Time-Frequency Analyses Bias Our Perception of EEG/MEG Signals and What to Do About It. *Frontiers in human neuroscience*, 13, 212. <https://doi.org/10.3389/fnhum.2019.00212>

Buzsáki, G., Anastassiou, C. A., & Koch, C. (2012). The origin of extracellular fields and currents--EEG, ECoG, LFP and spikes. *Nature reviews. Neuroscience*, 13(6), 407–420. <https://doi.org/10.1038/nrn3241>

Canolty, R. T., Edwards, E., Dalal, S. S., Soltani, M., Nagarajan, S. S., Kirsch, H. E., Berger, M. S., Barbaro, N. M., & Knight, R. T. (2006). High gamma power is phase-locked to theta oscillations in human neocortex. *Science (New York, N.Y.)*, 313(5793), 1626–1628. <https://doi.org/10.1126/science.1128115>

Gueguen, M. C. M., Lopez-Persem, A., Billeke, P., Lachaux, J. P., Rheims, S., Kahane, P., Minotti, L., David, O., Pessiglione, M., & Bastin, J. (2021). Anatomical dissociation of intracerebral signals for reward and punishment prediction errors in humans. *Nature communications*, 12(1), 3344. <https://doi.org/10.1038/s41467-021-23704-w>

Kiebel, S. J., Tallon-Baudry, C., & Friston, K. J. (2005). Parametric analysis of oscillatory activity as measured with EEG/MEG. *Human brain mapping*, 26(3), 170–177. <https://doi.org/10.1002/hbm.20153>

Knudsen, E. B., & Wallis, J. D. (2020). Closed-Loop Theta Stimulation in the Orbitofrontal Cortex Prevents Reward-Based Learning. *Neuron*, *106*(3), 537–547.e4. <https://doi.org/10.1016/j.neuron.2020.02.003>

Lachaux, J. P., Axmacher, N., Mormann, F., Halgren, E., & Crone, N. E. (2012). High-frequency neural activity and human cognition: past, present and possible future of intracranial EEG research. *Progress in neurobiology*, *98*(3), 279–301. <https://doi.org/10.1016/j.pneurobio.2012.06.008>

Lopez-Persem, A., Bastin, J., Petton, M., Abitbol, R., Lehongre, K., Adam, C., Navarro, V., Rheims, S., Kahane, P., Domenech, P., & Pessiglione, M. (2020). Four core properties of the human brain valuation system demonstrated in intracranial signals. *Nature neuroscience*, *23*(5), 664–675. <https://doi.org/10.1038/s41593-020-0615-9>

Mansouri, F. A., Freedman, D. J., & Buckley, M. J. (2020). Emergence of abstract rules in the primate brain. *Nature reviews. Neuroscience*, *21*(11), 595–610. <https://doi.org/10.1038/s41583-020-0364-5>

Manssuer, L., Qiong, D., Wei, L., Yang, R., Zhang, C., Zhao, Y., Sun, B., Zhan, S., & Voon, V. (2022). Integrated Amygdala, Orbitofrontal and Hippocampal Contributions to Reward and Loss Coding Revealed with Human Intracranial EEG. *The Journal of neuroscience : the official journal of the Society for Neuroscience*, *42*(13), 2756–2771. <https://doi.org/10.1523/JNEUROSCI.1717-21.2022>

Manssuer, L., Ding, Q., Zhang, Y., Gong, H., Liu, W., Yang, R., Zhang, C., Zhao, Y., Pan, Y., Zhan, S., Li, D., Sun, B., & Voon, V. (2023). Risk and aversion coding in human habenula high gamma activity. *Brain : a journal of neurology*, *146*(6), 2642–2653. <https://doi.org/10.1093/brain/awac456>

Saez, I., Lin, J., Stolk, A., Chang, E., Parvizi, J., Schalk, G., Knight, R. T., & Hsu, M. (2018). Encoding of Multiple Reward-Related Computations in Transient and Sustained High-Frequency Activity in Human OFC. *Current biology : CB*, *28*(18), 2889–2899.e3. <https://doi.org/10.1016/j.cub.2018.07.045>

Weber, J., Solbakk, A. K., Blenkmann, A. O., Llorens, A., Funderud, I., Leske, S., Larsson, P. G., Ivanovic, J., Knight, R. T., Endestad, T., & Helfrich, R. F. (2024). Ramping dynamics and theta oscillations reflect dissociable signatures during rule-guided human behavior. *Nature communications*, *15*(1), 637. <https://doi.org/10.1038/s41467-023-44571-7>

Xie, T., Adamek, M., Cho, H., Adamo, M. A., Ritaccio, A. L., Willie, J. T., Brunner, P., & Kubanek, J. (2024). Graded decisions in the human brain. *Nature communications*, *15*(1), 4308. <https://doi.org/10.1038/s41467-024-48342-w>